# A Non-monotonic Self-terminating Language Model

**Eugene Choi**[†]
eugene.choi@nyu.edu

**Kyunghyun Cho**[† ‡ §]
kyunghyun.cho@nyu.edu

**Cheolhyoung Lee**[† *]
cheolhyoung.lee@nyu.edu

## Abstract

Recent large-scale neural autoregressive sequence models have shown impressive performances on a variety of natural language generation tasks. However, their generated sequences often exhibit degenerate properties such as non-termination, undesirable repetition, and premature termination, when generated with decoding algorithms such as greedy search, beam search, top-$k$ sampling, and nucleus sampling. In this paper, we focus on the problem of non-terminating sequences resulting from an incomplete decoding algorithm. We first define an incomplete probable decoding algorithm which includes greedy search, top-$k$ sampling, and nucleus sampling, beyond the incomplete decoding algorithm originally put forward by Welleck et al. (2020). We then propose a non-monotonic self-terminating language model, which significantly relaxes the constraint of monotonically increasing termination probability in the originally proposed self-terminating language model by Welleck et al. (2020), to address the issue of non-terminating sequences when using incomplete probable decoding algorithms. We prove that our proposed model prevents non-terminating sequences when using not only incomplete probable decoding algorithms but also beam search. We empirically validate our model on sequence completion tasks with various architectures.

## 1 Introduction

Autoregressive neural sequence models (Bengio et al., 2000) have been widely used for various natural language generation tasks such as language modeling (Brown et al., 2020; Chowdhery et al., 2022), machine translation (Bahdanau et al., 2014), and conversational dialogue modeling (Vinyals & Le, 2015). Furthermore, large-scale autoregressive neural sequence models have shown unprecedented ability to generate fluent, human-like texts (Vaswani et al., 2017; Brown et al., 2020). Despite their success, the autoregressive neural sequence models have shown undesirable behaviors: non-termination (Welleck et al., 2020), degenerate repetition (Welleck et al., 2019; Holtzman et al., 2020), and premature termination (Koehn & Knowles, 2017; Stahlberg & Byrne, 2019). In this paper, we focus on how to prevent non-termination when using a given decoding algorithm.

*Non-termination* is the problem that a language model generates infinitely long sequences with a positive probability from our language model when using a given decoding algorithm. Welleck et al. (2020) pointed out that this issue comes from a discrepancy between the distribution of our language model and its induced distribution by an incomplete decoding algorithm. They formalized this disparity by the notion of *inconsistency* where our language model generates non-terminating sequences with a positive probability from the decoding algorithm. To avoid this inconsistency, they proposed a *self-terminating (ST) language model* that uses new parametrization for its classifier rather than usual softmax parametrization. They proved that the ST language model is consistent with respect to greedy search, beam search, top-$k$ sampling (Fan et al., 2018) as well as nucleus sampling (Holtzman et al., 2020).

The ST language model increases the termination probability of each sequence *monotonically* to 1, but this parametrization is not appropriate for learning our natural language. As an illustrative

---

[†]New York University

[‡]Prescient Design, Genentech

[§]CIFAR Fellow

[*]Corresponding author.

example, suppose there are two sequences in our dataset: *"I am a boy"* vs. *"I am a boy, and you are a girl."*. Our language model trained on this dataset may or may not terminate after the former. Once our model decides not to end, it should dramatically reduce the termination probability to continue. The ST language model, which monotonically increase the termination probability, cannot capture such a case, where one sequence is a prefix of another. We thus propose a *non-monotonic self-terminating (NMST)* language model which guarantees the consistency with respect to greedy search, beam search, top-$k$ sampling, and nucleus sampling without monotonically increasing termination probability.

The NMST language model encourages the termination probability of each sequence to converge to 1 through NMST parametrization however without monotonicity. Even under this relaxation, the proposed NMST language model provably prevents any non-terminating sequence resulting from greedy search, beam search, top-$k$ sampling, and nucleus sampling, which we refer to as *incomplete probable decoding algorithms*.

We conduct experiments validating the effectiveness of our NMST language models on sequence completion tasks, as was done in earlier studies. We test NMST parametrization with various architectures. Specifically, we train RNN (Elman, 1990) and LSTM (Hochreiter & Schmidhuber, 1997) on WikiText-2 (Merity et al., 2016). We additionally finetune GPT-2 (Radford et al., 2019) on WikiText-103 (Merity et al., 2016). Across all these setups, NMST parametrization effectively prevents non-terminating sequences, especially when compared to softmax parametrization. Furthermore, we see that our NMST parametrization has better (lower) perplexities than those of ST parametrization, confirming the importance of relaxing the monotonic termination probability.

## 2 Notations and Background

### 2.1 Notations for Autoregressive Neural Sequence Models

**Sequences, vocabulary, and** $\langle eos \rangle$ We view an instance (e.g., a sentence and a paragraph) as a *sequence* $\boldsymbol{y} = (y_1, y_2, \ldots, y_T)$, where each $y_t$ is an element from a pre-defined finite set of discrete tokens, referred to as a *vocabulary* $\mathcal{V}$. $\mathcal{V}$ includes a special symbol $\langle eos \rangle$ that only appears at the *end of the sequence*. Every sequence $\boldsymbol{y}$ must end with $\langle eos \rangle$. We write the length of $\boldsymbol{y}$ as $|\boldsymbol{y}|$, and $y_{|\boldsymbol{y}|} = \langle eos \rangle$. We call $\boldsymbol{y}$ a *non-terminating sequence*, $|\boldsymbol{y}| = \infty$, if $y_t \neq \langle eos \rangle$ for all $t$.

**Embedding vectors** Each token $v \in \mathcal{V}$ is not a numerical vector so that we use an *embedding vector* $\boldsymbol{u}_v \in \mathbb{R}^m$ to represent $v$. To capture the notion of similarity between discrete tokens efficiently, we use an embedding vector $\boldsymbol{u}_v \in \mathbb{R}^m$ to project $v$ into continuous embedding space (Bengio et al., 2000; Mikolov et al., 2013b;a; Levy & Goldberg, 2014).

**Autoregressive neural sequence models** Bengio et al. (2000) proposed an *autoregressive neural sequence model* parametrized by $\boldsymbol{\theta} \in \mathbb{R}^k$. They factorized $p_{\boldsymbol{\theta}}(\boldsymbol{y}|\boldsymbol{x})$ into a product of the conditional probability of each token given all the previous tokens and an input in a predefined order as follows:

$$p_{\boldsymbol{\theta}}(\boldsymbol{y}|\boldsymbol{x}) = \prod_{t=1}^{T} p_{\boldsymbol{\theta}}(\boldsymbol{y}_t|\boldsymbol{y}_{<t}, \boldsymbol{x}), \tag{1}$$

where $\boldsymbol{y}_{<t}$ is a *t-prefix* of $\boldsymbol{y}$ and $\boldsymbol{x}$ is an input referred to as a *context*. For example, $\boldsymbol{x}$ represents either a prompt in sequence completion or a source-side sequence in machine translation.

There are several popular architectures for $p_{\boldsymbol{\theta}}$ such as RNN (Elman, 1990), LSTM (Hochreiter & Schmidhuber, 1997), GRU (Cho et al., 2014), and Transformer (Vaswani et al., 2017). As shown in equation 2, all these models utilize softmax classifiers. In this paper, we modify the parametrization of their softmax classifiers to prevent non-terminating sequences. We thus write a *vanilla language model*, regardless of its choice of architecture, that uses the original softmax parametrization as $p_{\boldsymbol{\theta}}^{va}$ defined in Definition 1.

**Definition 1.** *A vanilla language model* $p_{\boldsymbol{\theta}}^{va}$ *computes the conditional probability of each token given a* $t$-*prefix* $\boldsymbol{y}_{<t}$ *and a context* $\boldsymbol{x}$ *at each time step* $t$ *as follows:*

$$p_{\boldsymbol{\theta}}^{va}(y_t = v|\boldsymbol{y}_{<t}, \boldsymbol{x}) = \exp(\boldsymbol{u}_v^\top \boldsymbol{h}_t) / \sum_{v' \in \mathcal{V}} \exp(\boldsymbol{u}_{v'}^\top \boldsymbol{h}_t), \tag{2}$$

*where* $\boldsymbol{h}_t = f_{\boldsymbol{\theta}}(\boldsymbol{y}_t, \boldsymbol{h}_{t-1})$ *with* $\boldsymbol{h}_0 = \boldsymbol{0}$.[1]

---

[1]This definition stands for RNN, LSTM, and GRU. For Transformer, $\boldsymbol{h}_t = f_{\boldsymbol{\theta}}(\boldsymbol{y}_t, \boldsymbol{h}_{1:(t-1)})$.

**Training** For a given dataset, $\mathcal{D} = \left\{ \left( \boldsymbol{x}^{(n)}, \boldsymbol{y}^{(n)} \right) \right\}_{n=1}^N$, we maximize the joint probability assigned to the sequences in our training dataset to find an optimal parameter configuration $\boldsymbol{\theta}^\star$ as follows:

$$\boldsymbol{\theta}^\star = \arg\max_{\boldsymbol{\theta}} \sum_{n=1}^N \sum_{t=1}^{T^{(n)}} \log p_{\boldsymbol{\theta}} \left( \boldsymbol{y}_t^{(n)} \big| \boldsymbol{y}_{<t}^{(n)}, \boldsymbol{x}^{(n)} \right). \tag{3}$$

## 2.2 Incomplete Probable Decoding Algorithms

An autoregressive language model $p_{\boldsymbol{\theta}}$ predicts the likelihood of a sequence $\boldsymbol{y}$ given a context $\boldsymbol{x}$. Its autoregressive factorization in equation 1 requires a recursive process for every $t$ to infer. Hence, at inference time, we use a decoding algorithm, defined below, to generate sequences from $p_{\boldsymbol{\theta}}$.

**Definition 2.** *Let $\mathcal{Y}$ be a collection of $\boldsymbol{y}$ such that $\boldsymbol{y} = (y_1, y_2, \cdots, y_T)$ where $T \in \{1, 2, \cdots\}$ and $y_t \in \mathcal{V}$. A decoding algorithm $\mathcal{S}$ is a function that maps $p_{\boldsymbol{\theta}}$ to $q_{\mathcal{S}(p_{\boldsymbol{\theta}})}$ which is a probability distribution over $\mathcal{Y}$. A decoded sentence $\hat{\boldsymbol{y}}$ given $\boldsymbol{x}$ by $\mathcal{S}$ from $p_{\boldsymbol{\theta}}$ is a random sample from $q_{\mathcal{S}(p_{\boldsymbol{\theta}})}(\boldsymbol{y}|\boldsymbol{x})$.*

To generate a high quality sequence from $p_{\boldsymbol{\theta}}$, a decoding algorithm assumes that a higher quality sequence has a higher probability of $p_{\boldsymbol{\theta}}$ than others. For instance, maximum *a posteriori* (MAP) decoding algorithm $\mathcal{S}_{map}$ gives the most probable sequence $\boldsymbol{y}^\star$ given a context $\boldsymbol{x}$ from $p_{\boldsymbol{\theta}}$:

$$\boldsymbol{y}^\star = \arg\max_{\boldsymbol{y} \in \mathcal{Y}} p_\theta(\boldsymbol{y}|\boldsymbol{x}), \tag{4}$$

by setting $q_{\mathcal{S}_{map}(p_{\boldsymbol{\theta}})}(\boldsymbol{y} = \boldsymbol{y}^\star|\boldsymbol{x}) = 1$ and $q_{\mathcal{S}_{map}(p_{\boldsymbol{\theta}})}(\boldsymbol{y} = \boldsymbol{y}'|\boldsymbol{x}) = 0$ where $\boldsymbol{y}' \in \mathcal{Y} \setminus \{\boldsymbol{y}^\star\}$. Unfortunately, $\mathcal{S}_{map}$ is intractable since equation 4 requires an exhaustive search over the sequence space $\mathcal{Y}$. Hence, in practice, we utilize *incomplete probable decoding algorithms* defined as follows:

**Definition 3.** *A decoding algorithm $\mathcal{S}$ is incomplete and probable if there exists $\mathcal{V}_t \subsetneq \mathcal{V}$ such that*

$$\sum_{v \in \mathcal{V}_t} q_{\mathcal{S}(p_{\boldsymbol{\theta}})}(y_t = v|\boldsymbol{y}_{<t}, \boldsymbol{x}) = 1 \tag{5}$$

*and*

$$\min_{v \in \mathcal{V}_t} p_{\boldsymbol{\theta}}(y_t = v|\boldsymbol{y}_{<t}, \boldsymbol{x}) \geq \max_{v \in \mathcal{V} \setminus \mathcal{V}_t} p_{\boldsymbol{\theta}}(y_t = v|\boldsymbol{y}_{<t}, \boldsymbol{x}) \tag{6}$$

*for each $t$. Furthermore, for every $v \in \mathcal{V}_t$, $\mathcal{S}$ satisfies*

$$q_{\mathcal{S}(p_{\boldsymbol{\theta}})}(y_t = v|\boldsymbol{y}_{<t}, \boldsymbol{x}) \geq p_{\boldsymbol{\theta}}(y_t = v|\boldsymbol{y}_{<t}, \boldsymbol{x}). \tag{7}$$

At each $t$, an incomplete probable decoding algorithm $\mathcal{S}$ considers only a set of highly probable tokens, $\mathcal{V}_t$. $\mathcal{S}$ generates $\hat{\boldsymbol{y}}$ given $\boldsymbol{x}$ by recursively sampling $\hat{y}_t$ from $q_{\mathcal{S}(p_{\boldsymbol{\theta}})}(y_t|\hat{\boldsymbol{y}}_{<t}, \boldsymbol{x})$ supported on $\mathcal{V}_t$. This reduces an exponential complexity of $\mathcal{S}_{map}$, $\mathcal{O}\left(|\mathcal{V}|^{|\hat{\boldsymbol{y}}|}\right)$, down to a linear level, $\mathcal{O}\left(|\hat{\boldsymbol{y}}| \cdot |\mathcal{V}|\right)$.

Greedy search, top-$k$ sampling (Fan et al., 2018), and nucleus sampling (Holtzman et al., 2020) are incomplete and probable. For example, greedy search $\mathcal{S}_{gr}$ generates the $t$-th item of a sequence by

$$\hat{y}_t = \arg\max_{v \in \mathcal{V}} p_{\boldsymbol{\theta}}(y_t = v|\hat{\boldsymbol{y}}_{<t}, \boldsymbol{x}). \tag{8}$$

In other words, $\mathcal{S}_{gr}$ sets $\mathcal{V}_t$ to $\left\{ v_t^{(1)} \right\}$ where $v_t^{(1)} = \arg\max_{v \in \mathcal{V}} p_{\boldsymbol{\theta}}(y_t = v|\hat{\boldsymbol{y}}_{<t}, \boldsymbol{x})$. Moreover, we have $p_{\boldsymbol{\theta}}\left(y_t = v_t^{(1)} \big| \hat{\boldsymbol{y}}_{<t}, \boldsymbol{x}\right) \leq q_{\mathcal{S}_{gr}(p_{\boldsymbol{\theta}})}\left(y_t = v_t^{(1)} \big| \hat{\boldsymbol{y}}_{<t}, \boldsymbol{x}\right) = 1$, and $q_{\mathcal{S}_{gr}(p_{\boldsymbol{\theta}})}(y_t = v'|\hat{\boldsymbol{y}}_{<t}, \boldsymbol{x}) = 0$ holds for $v' \in \mathcal{V} \setminus \mathcal{V}_t$. Thus, $\mathcal{S}_{gr}$ is incomplete and probable. Unlike $\mathcal{S}_{gr}$, top-$k$ sampling considers $k$ most probable tokens in $\mathcal{V}$ as $\mathcal{V}_t$ while nucleus sampling sets the smallest subset of $\mathcal{V}$, containing most probable tokens of which total probability is higher than a given threshold $\mu$, to $\mathcal{V}_t$. In §A.1 and A.2, we present that top-$k$ sampling and nucleus sampling are also incomplete and probable.

Beam search is a heuristic algorithm that operates on the level of prefixes. We describe it further in §A.3. Although beam search is not an incomplete probable decoding algorithm, it also selects $\mathcal{V}_t$ which is a proper subset of $\mathcal{V}$ to expand each prefix at each step $t$. Due to this, our main theoretical finding for the incomplete probable decoding algorithms in §3 is applicable to beam search as well.

## 2.3 Consistency with respect to Incomplete Probable Decoding Algorithms and Self-terminating (ST) Language Models

Incomplete probable decoding algorithms greatly reduce computational overhead for generating sequences from our model. However, Welleck et al. (2020) observed that they can generate non-terminating sequences even if every training sequence has a finite length. To study this, Welleck et al. (2020) defined consistency with respect to decoding algorithms as shown in Definition 4.

**Definition 4.** *A language model $p_{\boldsymbol{\theta}}$ is consistent with respect to a decoding algorithm $\mathcal{S}$ if $q_{\mathcal{S}(p_{\boldsymbol{\theta}})}(|\boldsymbol{y}| = \infty) = 0$ for any parameter configuration $\boldsymbol{\theta} \in \mathbb{R}^k$.*

Welleck et al. (2020) also proved that a vanilla language model $p_{\boldsymbol{\theta}}^{va}$ defined in Definition 1 is inconsistent with respect to incomplete probable decoding algorithms and beam search as follows:

**Theorem 1.** *A vanilla language model $p_{\boldsymbol{\theta}}^{va}$ defined in Definition 1 is inconsistent with respect to any incomplete probable decoding algorithm and beam search (Theorem 3.4 in Welleck et al. (2020)).*

For each $t$, an incomplete probable decoding algorithm $\mathcal{S}$ selects $\mathcal{V}_t \subsetneq \mathcal{V}$ as a set of candidates for decoding, but $p_{\boldsymbol{\theta}}^{va}$ does not guarantee that $\langle eos \rangle \in \mathcal{V}_t$. Specifically, if $\langle eos \rangle \notin \mathcal{V}_t$ for all $t$, then $\mathcal{S}$ cannot decode each token to $\langle eos \rangle$ for all $t$ (i.e., non-terminating). Based on this result, Welleck et al. (2020) proposed a *self-terminating (ST) language model*, defined below:

**Definition 5.** *For $\boldsymbol{h}_t$ defined in Definition 1, the conditional probability of each token $v \in \mathcal{V}$ given a $t$-prefix $\boldsymbol{y}_{<t}$ and a context $\boldsymbol{x}$ at each time step $t$ in an ST language model is given by*

$$\alpha_t = p_{\boldsymbol{\theta}}^{st}(y_t = \langle eos \rangle | \boldsymbol{y}_{<t}, \boldsymbol{x}) = 1 - \prod_{t'=1}^{t}(1-\epsilon) \cdot \sigma(\boldsymbol{u}_{\langle eos \rangle}^{\top} \boldsymbol{h}_{t'}), \qquad (9)$$

*and*

$$p_{\boldsymbol{\theta}}^{st}(y_t = v | \boldsymbol{y}_{<t}, \boldsymbol{x}) = (1 - \alpha_t) \cdot \exp(\boldsymbol{u}_v^{\top} \boldsymbol{h}_t) / \textstyle\sum_{v' \in \mathcal{V} \setminus \{\langle eos \rangle\}} \exp(\boldsymbol{u}_{v'}^{\top} \boldsymbol{h}_t),$$

*where $v \in \mathcal{V} \setminus \{\langle eos \rangle\}$, $\epsilon \in (0, 1)$, and $\sigma(x) = (1 + \exp(-x))^{-1}$ is a sigmoid function.*

They proved that the ST language model is consistent with respect to any incomplete probable decoding algorithm and beam search, as follows:

**Theorem 2.** *An ST language model $p_{\boldsymbol{\theta}}^{st}$ defined in Definition 5 is consistent with respect to any incomplete probable decoding algorithms and beam search (Theorem 4.1-4.3 in Welleck et al. (2020)).*

In equation 9, $p_{\boldsymbol{\theta}}^{st}(y_t = \langle eos \rangle | \boldsymbol{y}_{<t}, \boldsymbol{x})$ monotonically increases to 1 as $t$ increases. $\mathcal{S}$ ends up including $\langle eos \rangle$ in $\mathcal{V}_t$ always for $t \geq t'$ with some $t'$, and $\lim_{t \to \infty} q_{\mathcal{S}(p_{\boldsymbol{\theta}})}(y_t = \langle eos \rangle | \boldsymbol{y}_{<t}, \boldsymbol{x}) = 1$ by equation 7. This guarantees $\mathcal{S}$ to terminate in a finite number of steps. Despite $p_{\boldsymbol{\theta}}^{st}$'s consistency, its validation perplexity degrades compared to $p_{\boldsymbol{\theta}}^{va}$ in sequence completion (Welleck et al., 2020). We suspect that such degradation comes from the core property of $p_{\boldsymbol{\theta}}^{st}$ that $p_{\boldsymbol{\theta}}(y_t = \langle eos \rangle | \boldsymbol{y}_{<t}, \boldsymbol{x})$ *monotonically* increases to 1 as $t$ increases. In Remark 1 below, we provide an example where the optimal $p_{\boldsymbol{\theta}^\star}(y_t = \langle eos \rangle | \boldsymbol{y}_{<t}, \boldsymbol{x})$ is not monotonic.

**Remark 1.** *Let $\mathcal{D} = \left\{ (\boldsymbol{x}^{(1)}, \boldsymbol{y}^{(1)}), (\boldsymbol{x}^{(2)}, \boldsymbol{y}^{(2)}) \right\}$ be a two-instance training dataset. Assume that there exists $t_0$ such that $\boldsymbol{y}_{<t_0} = \boldsymbol{y}_{<t_0}^{(1)} = \boldsymbol{y}_{<t_0}^{(2)}$. Suppose further that $t_0 = |\boldsymbol{y}^{(1)}| < |\boldsymbol{y}^{(2)}| - 1$ and $\boldsymbol{x} = \boldsymbol{x}^{(1)} = \boldsymbol{x}^{(2)}$. If $\boldsymbol{\theta}^\star$ is an optimal parameter configuration in equation 3 over $\mathcal{D}$. Then, $p_{\boldsymbol{\theta}^\star}\left(y_t^{(2)} = \langle eos \rangle | \boldsymbol{y}_{<t}^{(2)}, \boldsymbol{x}\right)$ is not monotonic with respect to $t$ (proved in §B).*

We can easily find such case in natural language satisfying the assumptions in Remark 1 by concatenating two sequences. We empirically demonstrate the existence of such cases in §4.2.

## 3 NON-MONOTONIC SELF-TERMINATING (NMST) LANGUAGE MODELS

The consistency of $p_{\boldsymbol{\theta}}^{st}$ comes from $\lim_{t \to \infty} p_{\boldsymbol{\theta}}^{st}(y_t = \langle eos \rangle | \boldsymbol{y}_{<t}, \boldsymbol{x}) = 1$, not the monotonically increasing $p_{\boldsymbol{\theta}}^{st}(y_t = \langle eos \rangle | \boldsymbol{y}_{<t}, \boldsymbol{x})$ as a function of $t$. This motivates us to propose a *non-monotonic self-terminating (NMST) language model* $p_{\boldsymbol{\theta}}^{nmst}$ that permits $p_{\boldsymbol{\theta}}^{nmst}(y_t = \langle eos \rangle | \boldsymbol{y}_{<t}, \boldsymbol{x})$ to be a non-monotonic function of $t$ while satisfying $\lim_{t \to \infty} p_{\boldsymbol{\theta}}^{nmst}(y_t = \langle eos \rangle | \boldsymbol{y}_{<t}, \boldsymbol{x}) = 1$ as follows:

**Definition 6.** *For $\boldsymbol{h}_t$ defined in Definition 1, the conditional probability of each token given a $t$-prefix $\boldsymbol{y}_{<t}$ and a context $\boldsymbol{x}$ at the $t$-th step in an NMST language model is defined by*

$$\alpha_t = p_{\boldsymbol{\theta}}^{nmst}(y_t = \langle eos \rangle | \boldsymbol{y}_{<t}, \boldsymbol{x}) = \left(1 - \sigma(\boldsymbol{u}_{\langle eos \rangle}^{\top} \boldsymbol{h}_t)\right)(1 - (1-\epsilon)^t) + \sigma(\boldsymbol{u}_{\langle eos \rangle}^{\top} \boldsymbol{h}_t), \qquad (10)$$

*and*

$$p_{\boldsymbol{\theta}}^{nmst}(y_t = v | \boldsymbol{y}_{<t}, \boldsymbol{x}) = (1 - \alpha_t) \cdot \exp(\boldsymbol{u}_v^{\top} \boldsymbol{h}_t) / \textstyle\sum_{v' \in \mathcal{V} \setminus \{\langle eos \rangle\}} \exp(\boldsymbol{u}_{v'}^{\top} \boldsymbol{h}_t),$$

*where $v \in \mathcal{V} \setminus \{\langle eos \rangle\}$, $\epsilon \in (0, 1)$, and $\sigma(x) = (1 + \exp(-x))^{-1}$ is a sigmoid function.*

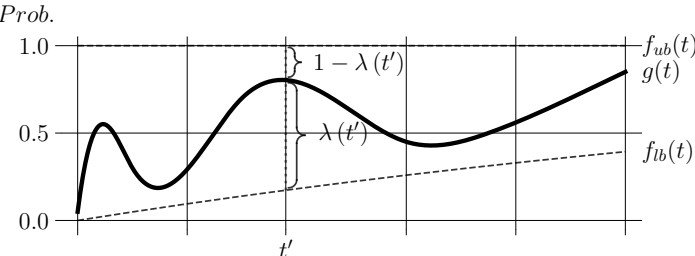

Figure 1: An illustration of NMST parametrization in equation 10 where $f_{lb}(t) = 1 - (1 - \epsilon)^t$, $f_{ub}(t) = 1$, $\lambda(t') = \sigma(\boldsymbol{u}_{\langle eos \rangle}^\top \boldsymbol{h}_{t'})$, and $g(t) = p_{\boldsymbol{\theta}}^{nmst}(y_t = \langle eos \rangle | \boldsymbol{y}_{<t}, \boldsymbol{x})$. If $g(t)$ lies between $f_{lb}(t)$ and $f_{ub}(t)$, we can find $\lambda(t')$ such that $g(t') = (1 - \lambda(t')) f_{lb}(t') + \lambda(t') f_{ub}(t')$ for any $t'$ regardless of whether $g(t)$ is monotonic with respect to $t$. This allows $p_{\boldsymbol{\theta}}^{nmst}$ to learn a non-monotonic behavior of $p_{\boldsymbol{\theta}}^{nmst}(y_t = \langle eos \rangle | \boldsymbol{y}_{<t}, \boldsymbol{x})$. $p_{\boldsymbol{\theta}}^{nmst}$ is consistent with respect to any incomplete probable decoding algorithms and beam search due to $\lim_{t \to \infty} f_{lb}(t) = 1 \Rightarrow \lim_{t \to \infty} p_{\boldsymbol{\theta}}^{nmst}(y_t = \langle eos \rangle | \boldsymbol{y}_{<t}, \boldsymbol{x}) = 1$.

Figure 1 shows that $p_{\boldsymbol{\theta}}^{nmst}$ uses convex combination of two curves to model $p_{\boldsymbol{\theta}}^{nmst}(y_t = \langle eos \rangle | \boldsymbol{y}_{<t}, \boldsymbol{x})$. We can write a curve $g(t)$ between a lower-bound curve $f_{lb}(t)$ and an upper-bound curve $f_{ub}(t)$ as $g(t) = (1 - \lambda(t)) f_{lb}(t) + \lambda(t) f_{ub}(t)$, with appropriate $\lambda(t) \in (0, 1)$ for all $t$. $p_{\boldsymbol{\theta}}^{nmst}$ sets $g(t)$ to $p_{\boldsymbol{\theta}}^{nmst}(y_t = \langle eos \rangle | \boldsymbol{y}_{<t}, \boldsymbol{x})$, and then regards it as a convex combination of $f_{lb}(t) = 1 - (1 - \epsilon)^t$ and $f_{ub}(t) = 1$ with a coefficient $\lambda(t) = \sigma(\boldsymbol{u}_{\langle eos \rangle}^\top \boldsymbol{h}_t)$. This enables non-monotonic $p_{\boldsymbol{\theta}}^{nmst}(y_t = \langle eos \rangle | \boldsymbol{y}_{<t}, \boldsymbol{x})$. Moreover, in Theorem 3 below, we show that the proposed NMST parametrization in equation 10 still guarantees the consistency with respect to any incomplete probable decoding algorithms and beam search.

**Theorem 3.** *An NMST language model defined in Definition 6 is consistent with respect to any incomplete probable decoding algorithms and beam search.*[2]

Theorem 3 guarantees that every decoded sequence from $p_{\boldsymbol{\theta}}^{nmst}$ terminates when using incomplete decoding algorithms and beam search. Neither $p_{\boldsymbol{\theta}}^{nmst}$ nor $p_{\boldsymbol{\theta}}^{st}$ results in non-terminating sequences resulting from incomplete probable decoding algorithms and beam search. Unlike ST parametrization, our NMST parametrization in equation 10 can capture a wider range of $p_{\boldsymbol{\theta}}(y_t = \langle eos \rangle | \boldsymbol{y}_{<t}, \boldsymbol{x})$, since $p_{\boldsymbol{\theta}}^{nmst}$ does not assume that $p_{\boldsymbol{\theta}}(y_t = \langle eos \rangle | \boldsymbol{y}_{<t}, \boldsymbol{x})$ is a monotonic function of $t$. We empirically demonstrate this by comparing $p_{\boldsymbol{\theta}}^{va}(y_t = \langle eos \rangle | \boldsymbol{y}_{<t}, \boldsymbol{x})$, $p_{\boldsymbol{\theta}}^{st}(y_t = \langle eos \rangle | \boldsymbol{y}_{<t}, \boldsymbol{x})$, and $p_{\boldsymbol{\theta}}^{nmst}(y_t = \langle eos \rangle | \boldsymbol{y}_{<t}, \boldsymbol{x})$ in Figure 3.

## 4 EXPERIMENTS

We empirically validate the effectiveness of the proposed non-monotonic self-terminating (NMST) language model by evaluating it on sequence completion tasks. We test three variants of a given architecture: (i) a vanilla (VA+) language model using common softmax parametrization in equation 2, (ii) a self-terminating (ST+) language model using ST parametrization proposed by Welleck et al. (2020) and (iii) our non-monotonic self-terminating (NMST+) language model using NMST parametrization in equation 10. We use following evaluation metrics for comparison:

- **Perplexity**: Given an autoregressive language model $p_{\boldsymbol{\theta}}$, the perplexity of $p_{\boldsymbol{\theta}}$ over $\mathcal{D}$ is
  $$\exp\left( -\frac{1}{N} \sum_{n=1}^{N} \sum_{t=1}^{T^{(n)}} \log p_{\boldsymbol{\theta}}\left( \boldsymbol{y}_t^{(n)} \Big| \boldsymbol{y}_{<t}^{(n)}, \boldsymbol{x}^{(n)} \right) \right), \text{ where } \mathcal{D} = \left\{ \left( \boldsymbol{x}^{(n)}, \boldsymbol{y}^{(n)} \right) \right\}_{n=1}^{N}.$$

- **Non-termination ratio** ($r_{nt}$): To present the consistency of $p_{\boldsymbol{\theta}}$ with respect to a given decoding algorithm $\mathcal{S}$, we need to compute $r_{nt} = q_{\mathcal{S}(p_{\boldsymbol{\theta}})}(|\boldsymbol{y}| = \infty)$. Instead, based on
  $$r_{nt} = q_{\mathcal{S}(p_{\boldsymbol{\theta}})}(|\boldsymbol{y}| = \infty) = \lim_{L \to \infty} q_{\mathcal{S}(p_{\boldsymbol{\theta}})}(|\boldsymbol{y}| > L), \tag{11}$$
  we use $r_{nt}(L) = q_{\mathcal{S}(p_{\boldsymbol{\theta}})}(|\boldsymbol{y}| > L)$ with a sufficiently large threshold $L$ to estimate $r_{nt}$.

Sequence completion is a task of predicting a continuation $\hat{\boldsymbol{y}}$ given a $c$-length context $\boldsymbol{x} = (x_1, x_2, \cdots, x_c)$ by using a decoding algorithm $\mathcal{S}$ from a language model $p_{\boldsymbol{\theta}}$ (i.e. $\hat{\boldsymbol{y}} \sim q_{\mathcal{S}(p_{\boldsymbol{\theta}})}(\boldsymbol{y}|\boldsymbol{x})$).

---

[2]We provide the proof in §C.

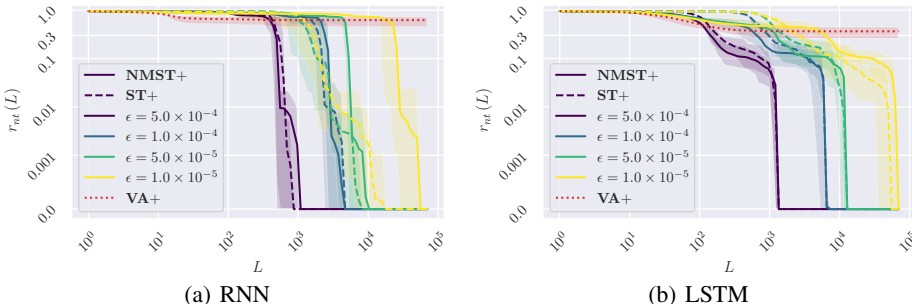

(a) RNN                       (b) LSTM

Figure 2: Non-termination ratios, $r_{nt}(L)$'s, as a function of $L$ in log-log scale for (a) RNN and (b) LSTM trained on WikiText-2 when using greedy search. We report mean (curve) $\pm$ st.dev. (shaded area) across 10 random experiments. For all configurations, both ST+ (non-red dashed) proposed by Welleck et al. (2020) and our NMST+ (non-red solid) are consistent with respect to greedy search since $r_{nt}(L)$ goes to 0 as $L$ increases. However, softmax parametrization (VA+, red dotted) is inconsistent with respect to greedy search since its $r_{nt}(L)$ does not converge to 0 as $L \to \infty$.

In this section, we use greedy search defined in equation 8 to generate $\hat{y}$ given $x$. Our main theoretical finding in Theorem 3 is that the proposed NMST language model is consistent with respect to not only greedy search but also top-$k$ sampling, nucleus sampling, and beam search. We thus present results when using decoding algorithms other than greedy search at the end in §5 and §F.

## 4.1 WIKITEXT-2

WikiText-2 (Merity et al., 2016) consists of 2 million words from 600 Wikipedia articles. With word tokenization, we regard the first 10 tokens of each sequence and its remaining part, as a context $x$ and a ground truth $y$, respectively. We train RNN with $\tanh$ (Elman, 1990) and LSTM (Hochreiter & Schmidhuber, 1997) on WikiText-2. Both RNN and LSTM have 2 layers, with 256 and 512 hidden units at each layer, respectively. We perform 10 random runs with a batch size of 32 for 70 epochs. We use AdamW (Loshchilov & Hutter, 2017) with an initial learning rate of 0.001, $\beta_1 = 0.9$, $\beta_2 = 0.99$, weight decay of 0.01, learning rate decay, and early stopping. We further describe our models and training strategies for WikiText-2 experiments in §D. Unlike VA+{RNN, LSTM}, ST+{RNN, LSTM} and NMST+{RNN, LSTM} need an additional hyperparameter $\epsilon$. We explore $\epsilon$ in $\{1.0 \times 10^{-5}, 5.0 \times 10^{-5}, 1.0 \times 10^{-4}, 5.0 \times 10^{-4}\}$.

We present the average ($\pm$st.dev.) non-termination ratios, $r_{nt}(L)$'s, across 10 random runs as a function of $L$ for all considered setups of WikiText-2 in Figure 2, using greedy search. From equation 11, a language model is consistent with respect to greedy search if $\lim_{L\to\infty} r_{nt}(L) = 0$. As $L$ increases, we observe that $r_{nt}(L)$'s of VA+{RNN, LSTM} fail to converge toward 0 while $r_{nt}(L)$'s of ST+{RNN, LSTM} and NMST+{RNN, LSTM} all reach 0. In other words, RNN and LSTM are now consistent with respect to greedy search after replacing the original softmax parametrization with either the proposed NMST parametrization or ST parametrization.

Table 1 shows that the average ($\pm$st.dev.) validation perplexities across 10 random experiments for all variants of RNN and LSTM, trained on WikiText-2. We observe that NMST+{RNN, LSTM} have better validation perplexities than ST+{RNN, LSTM} for every $\epsilon$. We demonstrate this more clearly in §E.1 by plotting the evolution of the mean validation perplexities as we vary $\epsilon$. Although our NMST+ guarantees the consistency of RNN and LSTM with respect to greedy search with a better validation perplexity than ST+, we need to carefully select $\epsilon$ of NMST+. As $\epsilon$ increases, the lower bound of $p_{\boldsymbol{\theta}}^{nmst}(y_t = \langle eos \rangle | \boldsymbol{y}_{<t}, \boldsymbol{x})$ grows faster, yielding premature sequences when $\epsilon$ is too large. Indeed, the average validation perplexities of NMST+RNN and NMST+LSTM with $\epsilon = 5.0 \times 10^{-4}$ are 184.2 and 105.6 which degrade by 5.6 and 4.0 from those of VA+RNN and VA+LSTM, 178.6 and 101.6, respectively. We however emphasize that there is the optimal $\epsilon = 1.0 \times 10^{-5}$ that makes NMST+{RNN, LSTM} have the validation perplexities similar to those of VA+{RNN, LSTM}. In short, both NMST+ and ST+ prevent non-termination when using greedy search but only NMST+ has a competitive validation perplexity against VA+. In §G, we further observe that the length distribution of predicted sequences from NMST+LSTM is closer to the length distribution of ground truth sequences than those of predicted sequences from {VA, ST}+LSTM.

Table 1: Mean ($\pm$st.dev.) validation perplexities across 10 random runs on WikiText-2 for various model configurations. Lower is better. **Bold** marks the best of each architecture. For all $\epsilon$, the validation perplexities of our NMST+{RNN, LSTM} are better than those of ST+{RNN, LSTM} proposed by Welleck et al. (2020). Moreover, with a proper choice of $\epsilon = 1.0 \times 10^{-5}$, NMST+{RNN, LSTM} have competitive validation perplexities against those of VA+{RNN, LSTM}.

| | RNN | | LSTM | |
|---|---|---|---|---|
| $\epsilon$ | ST+ | NMST+ | ST+ | NMST+ |
| $5.0 \times 10^{-4}$ | $186.1 \pm (6.2)$ | $184.2 \pm (6.5)$ | $106.1 \pm (1.0)$ | $105.6 \pm (1.2)$ |
| $1.0 \times 10^{-4}$ | $181.0 \pm (3.8)$ | $177.4 \pm (7.0)$ | $104.6 \pm (1.4)$ | $102.5 \pm (1.0)$ |
| $5.0 \times 10^{-5}$ | $182.6 \pm (8.0)$ | $179.6 \pm (5.7)$ | $104.7 \pm (1.6)$ | $102.1 \pm (1.0)$ |
| $1.0 \times 10^{-5}$ | $180.4 \pm (3.3)$ | $\mathbf{177.4 \pm (4.5)}$ | $104.5 \pm (1.4)$ | $\mathbf{101.5 \pm (0.8)}$ |
| VA+ | $178.6 \pm (6.3)$ | | $101.6 \pm (1.0)$ | |

Table 2: We present the average ($\pm$st.dev.) validation perplexities across 10 random runs for all variants of GPT-2 finetuned on WikiText-103. We also demonstrate their non-termination ratios (mean$\pm$st.dev.), $r_{nt}(L)$'s, when using greedy search. We set $L$ to 1,000 since the maximum length of generated sequences from GPT-2 is 1,024. For perplexity, lower is better. **Bold** marks the best validation perplexity in all setups. For every $\epsilon$, NMST+GPT-2 outperforms ST+GPT-2 in terms of the average validation perplexity. From $r_{nt}(L)$, NMST+GPT-2 effectively prevents non-termination sequences compared to VA+GPT-2 for every $\epsilon$ while ST+GPT-2 with small $\epsilon$ fails to avoid them. With a proper choice of $\epsilon$ (e.g., $\epsilon = 1.0 \times 10^{-5}$), NMST+GPT-2 improves the validation perplexity.

| | Perplexity | | $r_{nt}(L)$ | |
|---|---|---|---|---|
| $\epsilon$ | ST+ | NMST+ | ST+ | NMST+ |
| $5.0 \times 10^{-4}$ | $21.80 \pm (0.02)$ | $21.63 \pm (0.02)$ | $0.05 \pm (0.03)$ | $0.07 \pm (0.03)$ |
| $1.0 \times 10^{-4}$ | $21.21 \pm (0.02)$ | $20.86 \pm (0.02)$ | $0.72 \pm (0.11)$ | $0.22 \pm (0.10)$ |
| $5.0 \times 10^{-5}$ | $21.19 \pm (0.03)$ | $20.76 \pm (0.02)$ | $0.72 \pm (0.11)$ | $0.24 \pm (0.10)$ |
| $1.0 \times 10^{-5}$ | $21.16 \pm (0.03)$ | $\mathbf{20.69 \pm (0.03)}$ | $0.75 \pm (0.10)$ | $0.23 \pm (0.10)$ |
| VA+ | $20.72 \pm (0.03)$ | | $0.27 \pm (0.08)$ | |

## 4.2 WIKITEXT-103

WikiText-103 (Merity et al., 2016) consists of 103 million words constructed from 28,000 articles. We use BPE tokenization (Sennrich et al., 2015) and consider the first 10 tokens as a context for each sequence. Since WikiText-103 is substantially larger than WikiText-2, we finetune a pretrained GPT-2 which is a transformer language model with 124 million parameters (Radford et al., 2019) for $500,000$ steps. For computational efficiency, we bucket the dataset into sequences of similar lengths, and each batch contains a maximum of 1,024 total tokens. We use AdamW (Loshchilov & Hutter, 2017) with an initial learning rate of $5.0 \times 10^{-5}$, $\beta_1 = 0.9$, $\beta_2 = 0.99$, weight decay of 0.01, linear learning rate decay, and early stopping. We present a more detailed description in §D. We select $\epsilon$ from $\{1.0 \times 10^{-5}, 5.0 \times 10^{-5}, 1.0 \times 10^{-4}, 5.0 \times 10^{-4}\}$ for ST+GPT-2 and NMST+GPT-2.

We report the mean ($\pm$st.dev.) validation perplexities and non-termination ratios, $r_{nt}(L)$'s, resulting from greedy search across 10 random runs for all GPT-2 setups finetuned on WikiText-103 in Table 2. Since GPT-2 can handle up to 1,024 tokens, we use $L = 1,000$. As shown in Figure 2, we need a sufficiently large $L$ such as $L = 10^5$ to determine whether a language model is consistent with respect to greedy search. Although $L = 1,000$ is not sufficiently large, we observe that $r_{nt}(L)$ of NMST+GPT-2 decreases compared to $r_{nt}(L)$ of VA+GPT-2 as $\epsilon$ increases. That is, NMST+ reduces the number of non-terminating continuations within 1,000 steps. Non-terminating sequences do not necessarily imply better quality. We thus demonstrate sample continuations from NMST+GPT-2, given a context that leads non-termination with VA+GPT-2 in Table 3, using greedy search. We observe that the quality of the generated sequence tends to improve with NMST+ by avoiding repetitions of similar phrases and ending with $\langle eos \rangle$. We present more example continuations in §E.3.

Table 3: Given a context in a validation instance of WikiText-103, we present example continuations of {VA, ST, NMST}+GPT-2 when using greedy search. We select $\epsilon = 1.0 \times 10^{-5}$ for {ST, NMST}+GPT-2 because it is optimal in terms of validation perplexities in Table 2. Unlike {VA, ST}+GPT-2, NMST+GPT-2 improves the quality of the sequence by avoiding repetitive tokens and ending with $\langle eos \rangle$ when the given context leads VA+GPT-2 to non-terminate within $1,000$ steps.

| Context | *Made of concrete, steel, and wood, the* |
|---|---|
| VA+ | building was built in the mid @-@ 19th century. It was the first building in the United States to be built in concrete, and the first to be built in wood. It was also the first building in the United States to be built in steel. It was the first building in ... |
| ST+ | building is constructed of steel and concrete. The building's exterior is made of steel and concrete. The building's interior is made of wood, and the building's exterior is made of concrete. The building's exterior is made of concrete, and the building's ... |
| NMST+ | building was designed by the architectural firm of Bowers & Wainwright, and was completed in 1892. The building is the largest of its kind in the United States. $\langle eos \rangle$ |

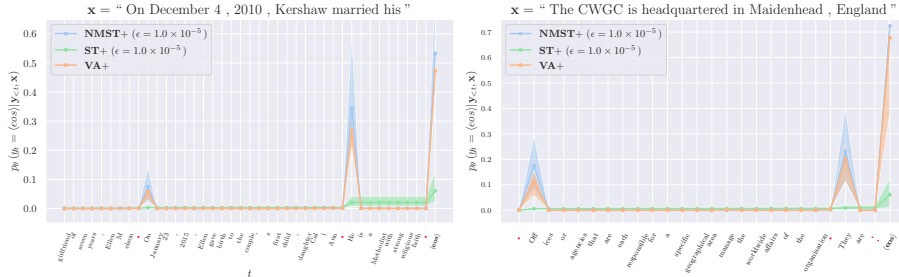

Figure 3: We present $p_\theta(y_t = \langle eos \rangle | \boldsymbol{y}_{<t}, \boldsymbol{x})$ as a function of $t$ for validation instances of WikiText-103 where $p_{\boldsymbol{\theta}}$'s are {VA, ST, NMST}+GPT-2. For {ST, NMST}+GPT-2, we choose $\epsilon = 1.0 \times 10^{-5}$ because it is optimal in terms of validation perplexities in Table 2. Instead of $t$, we tag the $t$-th ground truth token. We report their mean (curve) $\pm$ st.dev. (shaded area) across 10 random runs. Unlike ST+GPT-2, NMST+GPT-2 can model non-monotonic behaviors of $p_\theta(y_t = \langle eos \rangle | \boldsymbol{y}_{<t}, \boldsymbol{x})$ with respect to $t$. Both plots show that the non-monotonic behaviors occur where the sequences could end (e.g., after red marked tokens such as periods).

Similar to the results in §4.1, Table 2 shows that the validation perplexities of both ST+GPT-2 proposed by Welleck et al. (2020) and our NMST+GPT-2 degrade compared to VA+GPT-2 as $\epsilon$ increases. NMST+GPT-2 with the optimal $\epsilon = 1.0 \times 10^{-5}$ has a competitive validation perplexity of 20.69 against that of VA+GPT-2, 20.72. On the other side, we cannot find $\epsilon$ that makes the validation perplexity of ST+GPT-2 competitive against that of VA+GPT-2. Moreover, if $\epsilon \neq 5.0 \times 10^{-4}$, then $r_{nt}(L)$'s of ST+GPT-2 blow up unlike $r_{nt}(L)$'s of VA+GPT-2. §E.2 demonstrates the inevitable perplexity degradation and exploding $r_{nt}(L)$ of ST+GPT-2. We suspect that it is due to monotonically increasing $p_{\boldsymbol{\theta}}(y_t = \langle eos \rangle | \boldsymbol{y}_{<t}, \boldsymbol{x})$ with respect to $t$.

We investigate behaviors of $p_{\boldsymbol{\theta}}(y_t = \langle eos \rangle | \boldsymbol{y}_{<t}, \boldsymbol{x})$ where $p_{\boldsymbol{\theta}}$'s are {VA, ST, NMST}+GPT-2 in Figure 3. Based on Table 2, we select the optimal $\epsilon = 1.0 \times 10^{-5}$ in terms of validation perplexities for {ST, NMST}+GPT-2. In Figure 3, {VA, NMST}+GPT-2 well-capture whether a sequence might end (e.g., after periods) by showing non-monotonic behaviors at those seemingly-terminating steps, but ST+GPT-2 cannot model such non-monotonic behaviors because it assumes that $p_{\boldsymbol{\theta}}(y_t = \langle eos \rangle | \boldsymbol{y}_{<t}, \boldsymbol{x})$ is a monotonic function of $t$. This constraint makes ST+GPT-2 generate often finite but unnecessarily long sequences with greedy search (i.e., higher $r_{nt}(L)$ than VA+GPT-2 for small $L$, but $r_{nt}(L) = 0$ for sufficiently large $L$). We demonstrate more behaviors in §E.4.

## 5 CONSISTENCY WITH RESPECT TO OTHER DECODING ALGORITHMS

We explore the effectiveness of our proposed non-monotonic self-terminating (NMST) language model when using decoding algorithms other than greedy search, such as top-$k$ sampling (Fan et al.,

Table 4: Mean ($\pm$st.dev.) non-termination ratios, $r_{nt}(L)$'s, across 10 random runs for the variants of GPT-2 finetuned on WikiText-103 with various decoding algorithms. We set $L$ to 1,000 due to GPT-2's context window size of 1,024. We use the optimal $\epsilon = 1.0 \times 10^{-5}$ in terms of average validation perplexities in Table 2 for both NMST+GPT-2 and ST+GPT-2. **Bold** marks the lowest $r_{nt}(L)$ within each decoding algorithm (column). Similar to greedy search in Table 2, for all decoding algorithms, $r_{nt}(L)$'s of NMST+GPT-2 are lower than those of ST+GPT-2 and VA+GPT-2. It means that NMST+ reduce the number of non-terminating sequences within 1,000 decoding steps.

|  | top-2 | top-4 | nucleus-0.2 | nucleus-0.4 | beam-2 | beam-4 |
|---|---|---|---|---|---|---|
| VA+ | $0.0 \pm (0.0)$ | $0.0 \pm (0.0)$ | $0.25 \pm (0.08)$ | $0.14 \pm (0.05)$ | $0.05 \pm (0.02)$ | $0.03 \pm (0.01)$ |
| ST+ | $0.0 \pm (0.0)$ | $0.0 \pm (0.0)$ | $0.73 \pm (0.11)$ | $0.55 \pm (0.15)$ | $0.29 \pm (0.10)$ | $0.15 \pm (0.07)$ |
| NMST+ | $0.0 \pm (0.0)$ | $0.0 \pm (0.0)$ | $\mathbf{0.21} \pm (0.10)$ | $\mathbf{0.10} \pm (0.06)$ | $\mathbf{0.03} \pm (0.02)$ | $\mathbf{0.01} \pm (0.01)$ |

2018), nucleus sampling (Holtzman et al., 2020), and beam search. All experimental setups and notations are the same as Section §4. According to Theorem 3, the NMST language model is consistent with respect to any incomplete decoding algorithms (e.g., greedy search, top-$k$ sampling, and nucleus sampling) and beam search for all $\epsilon > 0$. To validate this, we use top-$\{2, 4\}$ sampling, nucleus-$\{0.2, 0.4\}$ sampling, and beam search with a width of $\{2, 4\}$ (beam-$\{2, 4\}$) to generate sequences from NMST+GPT-2 finetuned on WikiText-103 with $\epsilon = 1.0 \times 10^{-5}$. The choice of $\epsilon = 1.0 \times 10^{-5}$ is made based on the validation perplexities in Table 2. Since the validation perplexity does not depend on decoding algorithms, we focus on the average ($\pm$st.dev.) non-termination ratios, $r_{nt}(L)$'s, across 10 random runs with $L = 1,000$ for each decoding algorithm in Table 4. We also present $r_{nt}(L)$'s of VA+GPT-2 and ST+GPT-2 with $\epsilon = 1.0 \times 10^{-5}$ as baselines.

Table 4 shows that our NMST+GPT-2 has the lowest $r_{nt}(L)$ with $L = 1,000$ for all decoding algorithms compared to VA+GPT-2 and ST+GPT-2 proposed by (Welleck et al., 2020). In other words, NMST+ effectively prevent non-terminating sequences within 1,000 time steps regardless of decoding algorithms. Comparing with greedy search in Table 2 ($r_{nt}(L)$ when $\epsilon = 1.0 \times 10^{-5}$), we observe that $r_{nt}(L)$'s decrease for all setups. As we discussed in §2.3, non-terminating sequences originate from the choice of $\langle eos \rangle \notin \mathcal{V}_t \subsetneq \mathcal{V}$ for all $t$ where $\mathcal{V}$ is a vocabulary and $\mathcal{V}_t$ is the $t$-th proper subset of $\mathcal{V}$, considered by a decoding algorithm at the $t$-th step. The decoding algorithms other than greedy search are likely to have $\langle eos \rangle$ in $\mathcal{V}_t$ and have the lower $r_{nt}(L)$ since their $|\mathcal{V}_t|$ are greater than or equal to $|\mathcal{V}_t| = 1$ of greedy search for all $t$. In the case of top-$\{2, 4\}$ sampling, we obtain $r_{nt}(L) = 0.0$ for VA+GPT-2. Even without NMST+, VA+ can avoid non-terminating sequences if we choose a proper decoding algorithm. We however emphasize that NMST+GPT-2 with $\epsilon = 1.0 \times 10^{-5}$ has a competitive validation perplexity against VA+GPT-2 in Table 2 and that it is guaranteed to terminate regardless of the choice of a decoding algorithm. We also empirically demonstrate the consistency of NMST+{RNN, LSTM} trained on WikiText-2 with respect to other decoding algorithms in §F.

## 6 CONCLUSION

Non-termination is a degenerate behavior we often observe when generating text from a well-trained language model. To prevent this, Welleck et al. (2020) proposed a self-terminating language model that encourages the termination probability of each sequence, which is the conditional probability of $\langle eos \rangle$ given a $t$-prefix and a context, to monotonically increase toward 1 as $t$ increases. In this paper, we theoretically demonstrate that monotonically increasing termination probability of each sequence is not a necessary condition for avoiding non-terminating sequences. We then propose a non-monotonic self-terminating language model where the termination probability for each sequence converges to 1 but not monotonically. Our non-monotonic self-terminating language models successfully address the issue of non-termination and achieve perplexities that are comparable to vanilla language models and are better than the original self-terminating language models.

REPRODUCIBILITY STATEMENT

To ensure the reproducibility of our paper, we provide our code available at `https://github.com/nyu-dl/non-monotonic-self-terminating-lm`.

ACKNOWLEDGMENTS

This work was supported by 42dot, Hyundai Motor Company (under the project Uncertainty in Neural Sequence Modeling), Samsung Advanced Institute of Technology (under the project Next Generation Deep Learning: From Pattern Recognition to AI), and NSF Award 1922658 NRT-HDR: FUTURE Foundations, Translation, and Responsibility for Data Science. This work was supported in part through the NYU IT High Performance Computing resources, services, and staff expertise.

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

APPENDIX

## A DEFINITIONS OF COMMON DECODING ALGORITHMS AND THEIR CHARACTERISTICS

In this section, we present mathematical definitions of top-$k$ sampling (Fan et al., 2018), nucleus sampling (Holtzman et al., 2020), greedy search, and beam search. We then demonstrate whether they are incomplete probable decoding algorithms.

### A.1 TOP-K SAMPLING

At each step $t$, top-$k$ sampling selects a subset of $k$ most probable tokens in a vocabulary $\mathcal{V}$. Top-$k$ sampling generates decoded sequences from a language model $p_{\boldsymbol{\theta}}$ as follows:

**Definition A.1** (Top-$k$ sampling (Fan et al., 2018)). *Top-$k$ sampling $\mathcal{S}_{top\text{-}k}$ generates a sequence from a language model $p_{\boldsymbol{\theta}}$ given a context $\boldsymbol{x}$ by recursively sampling $\hat{y}_t$ from*

$$q_{\mathcal{S}_{top\text{-}k}(p_{\boldsymbol{\theta}})}(y_t = v|\hat{\boldsymbol{y}}_{<t}, \boldsymbol{x}) = \begin{cases} \dfrac{p_{\boldsymbol{\theta}}(y_t = v|\hat{\boldsymbol{y}}_{<t}, \boldsymbol{x})}{\sum_{v' \in \mathcal{V}_t} p_{\boldsymbol{\theta}}(y_t = v'|\hat{\boldsymbol{y}}_{<t}, \boldsymbol{x})}, & \text{if } v \in \mathcal{V}_t, \\ 0, & \text{otherwise,} \end{cases} \tag{12}$$

*where*

$$\mathcal{V}_t = \arg \underset{v \in \mathcal{V}}{top\text{-}k} \, p_{\boldsymbol{\theta}}(y_t = v|\hat{\boldsymbol{y}}_{<t}, \boldsymbol{x}). \tag{13}$$

Except the trivial case $k = |\mathcal{V}|$, we have $\emptyset \subsetneq \mathcal{V}_t \subsetneq \mathcal{V}$ for all $t$. By equation 13, equation 6 holds. From equation 12, we see that top-$k$ sampling satisfies equation 5 and equation 7. Therefore, top-$k$ sampling is an incomplete probable decoding algorithm.

### A.2 NUCLEUS SAMPLING

At each step $t$, nucleus sampling selects the smallest subset of most probable tokens in a vocabulary $\mathcal{V}$, of which total probability is higher than a given threshold $\mu$. Nucleus sampling generates decoded sequences from a language model $p_{\boldsymbol{\theta}}$ as follows:

**Definition A.2** (Nucleus sampling (Holtzman et al., 2020)). *Nucleus sampling $\mathcal{S}_{nuc\text{-}\mu}$ generates a sequence from a language model $p_{\boldsymbol{\theta}}$ given a context $\boldsymbol{x}$ by recursively sampling $\hat{y}_t$ from*

$$q_{\mathcal{S}_{nuc\text{-}\mu}(p_{\boldsymbol{\theta}})}(y_t = v|\hat{\boldsymbol{y}}_{<t}, \boldsymbol{x}) = \begin{cases} \dfrac{p_{\boldsymbol{\theta}}(y_t = v|\hat{\boldsymbol{y}}_{<t}, \boldsymbol{x})}{\sum_{v' \in \mathcal{V}_t} p_{\boldsymbol{\theta}}(y_t = v'|\hat{\boldsymbol{y}}_{<t}, \boldsymbol{x})}, & \text{if } v \in \mathcal{V}_t, \\ 0, & \text{otherwise,} \end{cases} \tag{14}$$

*where $\mathcal{V}_t$ is the smallest subset of $\mathcal{V}$ such that*

$$\sum_{v \in \mathcal{V}_t} p_{\boldsymbol{\theta}}(y_t = v|\hat{\boldsymbol{y}}_{<t}, \boldsymbol{x}) \geq \mu. \tag{15}$$

If $\min_{v \in \mathcal{V}} p_{\boldsymbol{\theta}}(y_t = v|\boldsymbol{y}_{<t}, \boldsymbol{x}) \leq 1 - \mu$ for any context $\boldsymbol{x}$ and any $t$-prefix $\boldsymbol{y}_{<t}$, then we have $\emptyset \subsetneq \mathcal{V}_t \subsetneq \mathcal{V}$ for all $t$. Suppose that equation 6 does not hold for nucleus sampling. Then, this contradicts to $\mathcal{V}_t$ is the smallest subset of $\mathcal{V}$, satisfying equation 15. From equation 14, we see that nucleus sampling satisfies equation 5 and equation 7. Therefore, nucleus sampling is incomplete and probable.

### A.3 BEAM SEARCH

Beam search is a heuristic algorithm that operates on the level of prefixes. We use the definition of beam search in Welleck et al. (2020).

**Definition A.3** (Beam search, Definition A.2 in Welleck et al. (2020)). *Beam search with a width (beam size) $k$, $\mathcal{S}_{beam\text{-}k}$, generates a sequence from a language model $p_{\boldsymbol{\theta}}$ by maintaining a set of $k$*

prefixes, $\mathcal{P}_t = \{\boldsymbol{\rho}^{(1)}(t), \boldsymbol{\rho}^{(2)}(t), \cdots, \boldsymbol{\rho}^{(k)}(t)\}$, at each time step $t$ where $\boldsymbol{\rho}^{(i)}(0)$ is an empty prefix for all $i$. At each step $t \in \{1, 2, \cdots\}$, beam search forms a set of $k \times k$ prefixes,

$$\tilde{\mathcal{P}}_t = \bigcup_{\boldsymbol{\rho} \in \mathcal{P}_{t-1}} \{\boldsymbol{\rho} \circ v | v \in \mathcal{V}_t(\boldsymbol{\rho})\}, \tag{16}$$

where $\boldsymbol{\rho} \circ v$ is concatenation and

$$\mathcal{V}_t(\boldsymbol{\rho}) = \arg\operatorname*{top-}k_{v \in \mathcal{V}} p_{\boldsymbol{\theta}}(y_t = v | \boldsymbol{\rho}, \boldsymbol{x}). \tag{17}$$

After forming $\tilde{\mathcal{P}}_t$, beam search selects a set of the $k$ highest scoring prefixes in $\tilde{\mathcal{P}}_t$,

$$\mathcal{P}_t = \arg\operatorname*{top-}k_{\boldsymbol{\rho} \in \tilde{\mathcal{P}}_t} s(\boldsymbol{\rho}), \tag{18}$$

where $s(\boldsymbol{\rho}) = \sum_{\tau=1}^t \log p_{\boldsymbol{\theta}}(y_\tau = \boldsymbol{\rho}_\tau | \boldsymbol{\rho}_{<\tau}, \boldsymbol{x})$. If $\boldsymbol{\rho} \in \mathcal{P}_t$ ends with $\langle eos \rangle$, then it does not expand further and is added to the final set $\mathcal{P}$. Beam search continues until $\mathcal{P}$ contains $k$ sequences ending with $\langle eos \rangle$. After that it returns the highest scoring sequence

$$\hat{\boldsymbol{y}} = \arg\max_{\boldsymbol{\rho} \in \mathcal{P}} s(\boldsymbol{\rho}). \tag{19}$$

Unlike greedy search, top-$k$ sampling, and nucleus sampling, beam search recursively expands $k$ sequences with at most $k$ different prefixes. Therefore, we cannot formalize beam search in token-level by using $q_{\mathcal{S}_{\text{beam-}k}}(y_t = v | \boldsymbol{y}_{<t}, \boldsymbol{x})$. However, in equation 17, the number of possible tokens at $t$ is at most $k \times k$. It means that $\mathcal{S}_{\text{beam-}k}$ may exclude $\langle eos \rangle$ at time $t$ if $k \le \sqrt{|\mathcal{V}| - 1}$. By using this, Welleck et al. (2020) proved that a vanilla language model $p_{\boldsymbol{\theta}}^{va}$ is inconsistent with respect to beam search as shown in Theorem 1.

## B  PROOFS FOR §2.3

**Remark 1.** *Let $\mathcal{D} = \{(\boldsymbol{x}^{(1)}, \boldsymbol{y}^{(1)}), (\boldsymbol{x}^{(2)}, \boldsymbol{y}^{(2)})\}$ be a two-instance training dataset. Assume that there exists $t_0$ such that $\boldsymbol{y}_{<t_0} = \boldsymbol{y}_{<t_0}^{(1)} = \boldsymbol{y}_{<t_0}^{(2)}$. Suppose further that $t_0 = |\boldsymbol{y}^{(1)}| < |\boldsymbol{y}^{(2)}| - 1$ and $\boldsymbol{x} = \boldsymbol{x}^{(1)} = \boldsymbol{x}^{(2)}$. If $\boldsymbol{\theta}^\star$ is an optimal parameter configuration in equation 3 over $\mathcal{D}$. Then, $p_{\boldsymbol{\theta}^\star}(y_t^{(2)} = \langle eos \rangle | \boldsymbol{y}_{<t}^{(2)}, \boldsymbol{x})$ is non-monotonic with respect to $t$.*

*Proof.* Since $\boldsymbol{\theta}^\star$ is an optimal parameter configuration that perfectly minimizes equation 3 and $t_0 < |\boldsymbol{y}^{(2)}| - 1$, we have

$$p_{\boldsymbol{\theta}^\star}(y_t^{(2)} = \langle eos \rangle | \boldsymbol{y}_{<t}^{(2)}, \boldsymbol{x}^{(2)}) = 0, \tag{20}$$

for $t < t_0$. Note that $t_0 = |\boldsymbol{y}^{(1)}| \Rightarrow \boldsymbol{y}_{t_0}^{(1)} = \langle eos \rangle$ and $t_0 < |\boldsymbol{y}^{(2)}| - 1 \Rightarrow \boldsymbol{y}_{t_0}^{(2)} \ne \langle eos \rangle$. From $\boldsymbol{x} = \boldsymbol{x}^{(1)} = \boldsymbol{x}^{(2)}$ and $\boldsymbol{y} = \boldsymbol{y}_{<t_0}^{(1)} = \boldsymbol{y}_{<t_0}^{(2)}$, we obtain

$$p_{\boldsymbol{\theta}^\star}(y_{t_0}^{(2)} = \langle eos \rangle | \boldsymbol{y}_{<t_0}^{(2)}, \boldsymbol{x}^{(2)}) = \frac{1}{2}. \tag{21}$$

Moreover, $t_0 < |\boldsymbol{y}^{(2)}| - 1$ implies that $\boldsymbol{y}_{t_0+1}^{(2)} \ne \langle eos \rangle$ which is equivalent to

$$p_{\boldsymbol{\theta}^\star}(y_{t_0+1}^{(2)} = \langle eos \rangle | \boldsymbol{y}_{<t_0+1}^{(2)}, \boldsymbol{x}^{(2)}) = 0. \tag{22}$$

From equation 20, equation 21, and equation 22, we see that $p_{\boldsymbol{\theta}^\star}(y_t^{(2)} = \langle eos \rangle | \boldsymbol{y}_{<t}^{(2)}, \boldsymbol{x})$ is non-monotonic with respect to $t$. $\qquad\square$

## C  PROOFS FOR §3

**Theorem 3.** *A non-monotonic self-terminating (NMST) language model defined in Definition 6 is consistent with respect to any incomplete probable decoding algorithms and beam search.*

*Proof.* From equation 10, for any $\boldsymbol{\theta} \in \mathbb{R}^k$, we have

$$\lim_{t \to \infty} p_{\boldsymbol{\theta}}^{nmst}(y_t = \langle eos \rangle | \boldsymbol{y}_{<t}, \boldsymbol{x}) = 1,$$

since $(1 - \epsilon)^t \to 0$ as $t \to \infty$ for $\epsilon \in (0, 1)$ and $\sigma\left(\boldsymbol{u}_{\langle eos \rangle}^\top \boldsymbol{h}_t\right) \in (0, 1)$ for any $t$. Hence, there exists $t_{1/2}$ such that

$$t \geq t_{1/2} \Rightarrow p_{\boldsymbol{\theta}}^{nmst}(y_t = \langle eos \rangle | \boldsymbol{y}_{<t}, \boldsymbol{x}) > \frac{1}{2}. \tag{23}$$

Let $\mathcal{S}$ be any incomplete probable decoding algorithm. From equation 6 and equation 7, $\langle eos \rangle \in \mathcal{V}_t$ and $q_{\mathcal{S}(p_{\boldsymbol{\theta}}^{nmst})}(y_t \neq \langle eos \rangle | \boldsymbol{y}_{<t}, \boldsymbol{x}) < \frac{1}{2}$ holds for any $t \geq t_{1/2}$. Therefore, we obtain

$$
\begin{aligned}
q_{\mathcal{S}(p_{\boldsymbol{\theta}}^{nmst})}(|\boldsymbol{y}| = \infty | \boldsymbol{x}) &= \prod_{t=1}^{\infty} q_{\mathcal{S}(p_{\boldsymbol{\theta}}^{nmst})}(y_t \neq \langle eos \rangle | \boldsymbol{y}_{<t}, \boldsymbol{x}) \\
&\leq \prod_{t=t_{1/2}}^{\infty} q_{\mathcal{S}(p_{\boldsymbol{\theta}}^{nmst})}(y_t \neq \langle eos \rangle | \boldsymbol{y}_{<t}, \boldsymbol{x}) \\
&< \prod_{t=t_{1/2}}^{\infty} \frac{1}{2} \to 0.
\end{aligned}
\tag{24}
$$

Taking expectation of equation 24 over $\boldsymbol{x}$, we finally have $q_{\mathcal{S}(p_{\boldsymbol{\theta}}^{nmst})}(|\boldsymbol{y}| = \infty) = 0$ for any $\mathcal{S}$. In other words, $p_{\boldsymbol{\theta}}^{nmst}$ is consistent with respect to any incomplete probable decoding algorithms.

In the case of beam search $\mathcal{S}_{\text{beam-}k}$ defined in §A.3, without loss of generality, there exists $\boldsymbol{\rho} \in \mathcal{P}_{t_{1/2}}$ such that $\boldsymbol{\rho}$ does not end with $\langle eos \rangle$. [3] Let $\mathcal{P}_{>t_{1/2}}(\boldsymbol{\rho})$ be a set of $k$ highest scoring sequences continued from $\boldsymbol{\rho}$ by $\mathcal{S}_{\text{beam-}k}$. From equation 23, we have

$$p_{\boldsymbol{\theta}}^{nmst}(\langle eos \rangle | \boldsymbol{\rho}, \boldsymbol{x}) > p_{\boldsymbol{\theta}}^{nmst}(v | \boldsymbol{\rho}, \boldsymbol{x})$$

for all $v \in \mathcal{V} \setminus \{\langle eos \rangle\}$. Hence, $\mathcal{V}_{t_{1/2}}(\boldsymbol{\rho})$ in equation 17 includes $\langle eos \rangle$. Let $\boldsymbol{z} = (z_1, z_2, \cdots, z_l)$ be any subsequence with $z_1 \neq \langle eos \rangle$. Then, we have

$$
\begin{aligned}
p_{\boldsymbol{\theta}}^{nmst}(\boldsymbol{\rho} \circ \boldsymbol{z} | \boldsymbol{\rho}, \boldsymbol{x}) &= \prod_{i=1}^{l} p_{\boldsymbol{\theta}}^{nmst}(z_i | \boldsymbol{\rho} \circ \boldsymbol{z}_{<i}, \boldsymbol{x}) \\
&\leq p_{\boldsymbol{\theta}}^{nmst}(z_1 | \boldsymbol{\rho}, \boldsymbol{x}) \\
&< p_{\boldsymbol{\theta}}^{nmst}(\langle eos \rangle | \boldsymbol{\rho}, \boldsymbol{x}) = p_{\boldsymbol{\theta}}^{nmst}(\boldsymbol{\rho} \circ \langle eos \rangle | \boldsymbol{\rho}, \boldsymbol{x}),
\end{aligned}
\tag{25}
$$

where $\circ$ is concatenation. Therefore, $\boldsymbol{\rho} \circ \langle eos \rangle = \arg\max_{\boldsymbol{\rho}' \in \mathcal{P}_{t_{1/2}}} s(\boldsymbol{\rho}')$ holds where $s(\boldsymbol{\rho}') = \sum_{\tau=1}^{t} \log p_{\boldsymbol{\theta}}^{nmst}(\rho_\tau' | \boldsymbol{\rho}_{<\tau}', \boldsymbol{x})$. That is, $\boldsymbol{\rho} \circ \langle eos \rangle$ is the highest scoring sequence starting with $\boldsymbol{\rho}$, and we have $\boldsymbol{\rho} \circ \langle eos \rangle \in \mathcal{P}(\boldsymbol{\rho})$.

For each $\boldsymbol{\rho}' \in \mathcal{P}_{>t_{1/2}}(\boldsymbol{\rho}) \setminus \{\boldsymbol{\rho} \circ \langle eos \rangle\}$, $\boldsymbol{\rho}'$ starts with $\boldsymbol{\rho} \circ v$ for $v \in \mathcal{V} \setminus \{\langle eos \rangle\}$. By the same argument, we add at least one sequence ending with $\langle eos \rangle$ to $\mathcal{P}_{>t_{1/2}}(\boldsymbol{\rho})$. It means that $\mathcal{P}_{>t_{1/2}}(\boldsymbol{\rho})$ has $k$ sequences ending with $\langle eos \rangle$ within $t_{1/2} + k$ steps. Note that the final set $\mathcal{P}$ satisfies

$$\mathcal{P} \subset \bigcup_{\boldsymbol{\rho} \in \mathcal{P}_{t_{1/2}}} \mathcal{P}_{>t_{1/2}}(\boldsymbol{\rho}). \tag{26}$$

Equation 26 implies that every sequence in $\mathcal{P}$ has the length of at most $t_{1/2} + k$. We thus obtain

$$q_{\mathcal{S}_{\text{beam-}k}(p_{\boldsymbol{\theta}}^{nmst})}(|\boldsymbol{y}| = \infty | \boldsymbol{x}) \leq q_{\mathcal{S}_{\text{beam-}k}(p_{\boldsymbol{\theta}}^{nmst})}(|\boldsymbol{y}| > t_{1/2} + k | \boldsymbol{x}) = 0. \tag{27}$$

Taking expectation of equation 27 over $\boldsymbol{x}$, we see that $q_{\mathcal{S}_{\text{beam-}k}(p_{\boldsymbol{\theta}}^{nmst})}(|\boldsymbol{y}| = \infty)$. That is, $p_{\boldsymbol{\theta}}^{nmst}$ is consistent with respect to beam search. $\qquad \square$

---

[3]If there is no such $\boldsymbol{\rho}$, all $k$ sequences in $\mathcal{P}_{t_{1/2}}$ end with $\langle eos \rangle$. It means that $\mathcal{S}_{\text{beam-}k}$ returns a finite sequence, so that $p_{\boldsymbol{\theta}}^{nmst}$ is consistent with respect to beam search.

## D    EXPERIMENTAL DETAILS

In this section, we describe our models and optimization processes used in §4.

**RNN and LSTM on WikiText-2**    We use word tokenization for WikiText-2. We train RNN with $\tanh$ activations (Elman, 1990) and LSTM (Hochreiter & Schmidhuber, 1997) on WikiText-2. Both RNN and LSTM have 2 layers. Each layer has 256 hidden units for RNN and 512 hidden units for LSTM. The sizes of input and output embedding layers are 256 and 512 for RNN and LSTM, respectively. We use weight tying to share the weights between the input and output embedding layers for both models. We apply dropout (Srivastava et al., 2014) with drop probabilities of 0.3 and 0.5 to RNN and LSTM accordingly. For each model, we perform 10 random runs with a batch size of 32 for 70 epochs. To maximize the log-likelihood presented in equation 3, we use AdamW (Loshchilov & Hutter, 2017) with an initial learning rate of 0.001, $\beta_1 = 0.9$, $\beta_2 = 0.99$, weight decay of 0.01, and learning rate decay which halves the learning rate if the validation perplexity does not improve for a training epoch. To avoid overfitting, we additionally use early stopping, which terminates training if the validation perplexity does not improve upon the best score attained so far for 10 consecutive epochs. In most cases, the training ends within 50 epochs.

**GPT-2 on WikiText-103**    We use BPE tokenization[4] (Sennrich et al., 2015) and the pretrained GPT-2[5] (Radford et al., 2019) with 124 million parameters, provided by `HuggingFace`. GPT-2 can handle up to 1,024 tokens. We apply dropout (Srivastava et al., 2014) with a drop probability of 0.1 to GPT-2. We finetune GPT-2 for 300,000 steps while ensuring that all runs continue for at least 250,000 steps. To minimize the number of padding tokens in every batch for computational efficiency, we bucket the dataset into sequences of similar lengths, and each batch contains a maximum of 1,024 total tokens. To maximize the log-likelihood function in equation 3, we use AdamW (Loshchilov & Hutter, 2017) with an initial learning rate of $5.0 \times 10^{-5}$, $\beta_1 = 0.9$, $\beta_2 = 0.99$, weight decay of 0.01, and linear learning rate decay over $500,000$ steps.

---

[4]`https://github.com/huggingface/tokenizers`
[5]`https://github.com/huggingface/transformers`

# E ADDITIONAL PLOTS AND TABLES FOR §4

In this section, we demonstrate additional plots and tables for §4.

## E.1 ADDITIONAL PLOTS FOR §4.1

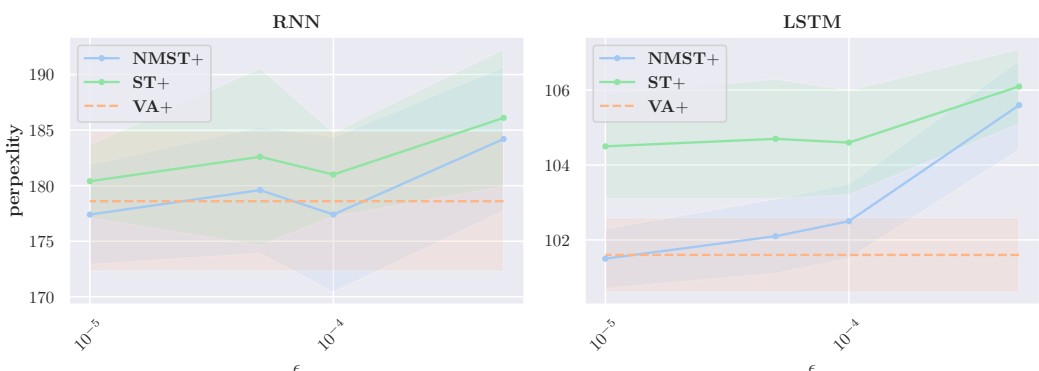

Figure 4: Validation perplexities as a function of $\epsilon$ in log-linear scale for all configurations of RNN (left) and LSTM (right), which are trained on WikiText-2. We present their average (curve) $\pm$ st.dev. (shaded area) across 10 random experiments. For all $\epsilon$ and architectures, NMST+ has better validation perplexities than ST+. As $\epsilon$ increases, the validation perplexities of both NMST+RNN and NMST+LSTM degrade compared to those of VA+RNN and VA+LSTM. We thus need to search for an optimal $\epsilon$ to avoid degradation of validation perplexity when applying NMST+ to our language model.

## E.2 ADDITIONAL PLOTS FOR §4.2

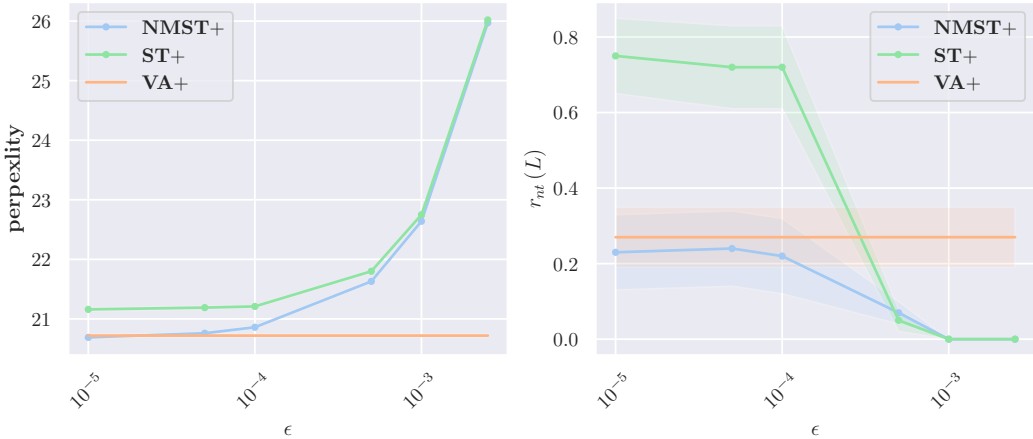

Figure 5: We present the average (curve) $\pm$ st.dev. (shaded area) of validation perplexities (left) and non-termnation ratios $r_{nt}(L)$ (right) with greedy search across 10 random runs for all considered setups of GPT-2 finetuned on WikiText-130 in log-linear scale. For $r_{nt}(L)$, we use $L = 1,000$ because GPT-2 has a context window size of $1,024$. For all $\epsilon$, NMST+GPT-2 outperforms ST+GPT-2 in terms of the average validation perplexity. When $\epsilon$ is small, $r_nt(L)$ of ST+GPT-2 explodes. It means that ST+GPT-2 with small $\epsilon$ cannot prevent non-terminating sequences. However, our NMST+GPT-2 effectively reduces $r_{nt}(L)$ compared to VA+GPT-2 for every $\epsilon$, and the validation perplexity degradation is smaller than that of ST+GPT-2 proposed by Welleck et al. (2020).

E.3    ADDITIONAL TABLES FOR TABLE 3

Table 5: Given a context in a validation instance of WikiText-103, we present example continuations of {VA, ST, NMST}+GPT-2 when using greedy search. We select $\epsilon = 1.0 \times 10^{-5}$ for {ST, NMST}+GPT-2 because it is optimal in terms of validation perplexities in Table 2. Unlike {VA, ST}+GPT-2, NMST+GPT-2 improves the quality of the sequence by avoiding repetitive tokens and ending with $\langle eos \rangle$ when the given context leads VA+GPT-2 to non-terminate within $1,000$ steps.

| Context | *The single made its Irish Singles Chart debut at* |
|---|---|
| VA+ | number three on the issue date of November 18, 2005. It debuted at number one on the chart for the issue date of December 7, 2005. The single debuted at number two on the chart for the issue date of January 4, 2006. The single debuted at number one on the Irish Singles Chart for the issue date of March 4, 2006. The single debuted at number two on the Irish Singles Chart for the issue date of June 4, 2006. The single debuted at number one on the Irish Singles Chart for the issue date ... |
| ST+ | number three on the chart issue dated March 16, 2010. It debuted at number two on the UK Singles Chart on the chart issue dated March 16, 2010, and peaked at number one on the chart issue dated March 16, 2010. The single was certified gold by the British Phonographic Industry ( BPI ) for shipments of over 15 @,@ 000 copies. The single debuted at number two on the Irish Singles Chart on the chart issue dated March 16, 2010, and peaked at number one on the chart issue dated March 16, 2010. The single was certified gold by the Irish Recorded Music Association ( IRMA ) for shipments of over 15 @,@ 000 copies. The single was ... |
| NMST+ | number twenty @-@ seven on the week ending March 26, 2010, and peaked at number three on the week ending March 27, 2010. It was certified gold by the Recording Industry Association of Ireland ( RIAA ) for shipment of 500 @,@ 000 copies of the single. The single was certified gold by the Recording Industry Association of Ireland ( RIANZ ) for shipment of 500 @,@ 000 copies of the single.$\langle eos \rangle$ |
| Context | *Despite the expensive reconstructions, both vessels were considered* |
| VA+ | to be of sufficient quality to be considered for use in the Grand Fleet. The first, the British @-@ built, British @-@ built, British @-@ built, British @-@ built, British @-@ built, British @-@ built, British @-@ built, British @-@ built, British @-@ built, British @-@ built, British @-@ built, British @-@ built, British @-@ built, British @-@ built, British @-@ built, British @-@ built, British @-@ built, British @-@ built, British @-@ built, British @-@ built, British @-@ built, British @-@ built, British @-@ built, British @-@ built, British @-@ built, British @-@ built, British @-@ built, British @-@ built, British @-@ built, British @-@ built, British @-@ built, British @-@ built, British @-@ built, British @-@ built, British @-@ built, British @-@ built, British @-@ built, British @-@ built, British @-@ built, British @-@ built, British @-@ built, British @-@ built, British @-@ built, British @-@ built, British @-@ built ... |
| ST+ | to be of sufficient quality to be considered a part of the Royal Navy, and were assigned to the Channel Fleet. The two ships were transferred to the Mediterranean Fleet in early 1915, and were transferred to the North America and West Indies Station in early 1916. They were transferred to the Mediterranean Fleet in early 1917, and were transferred to the North America and West Indies Station in early 1918. They were transferred to the North America and West Indies Station in early 1918, and were transferred to the Mediterranean Fleet in early 1918. They were transferred to the North America and West Indies Station in late 1918, and ... |
| NMST+ | obsolescent by the time of the Second World War, and were eventually scrapped in favour of the much more powerful British battleship HMS Dreadnought.$\langle eos \rangle$ |

| Context | *On the July 21, 2014 episode of Raw,* |
|---|---|
| VA+ | Triple H and Randy Orton were involved in a tag team match, in which Triple H and Orton were accompanied by Triple H and Randy Orton's manager, Mr. Kennedy. Orton was accompanied by Orton's manager, Mr. Kennedy, and Orton's tag team partner, Mr. Kennedy. Triple H and Orton were accompanied by Orton's manager, Mr. Kennedy, and Orton's tag team partner, Mr. Kennedy. Triple H and Orton were accompanied by Orton's manager, Mr. Kennedy, and Orton's manager, Mr. Kennedy. Triple H and Orton were accompanied by Orton's manager, Mr. Kennedy, and Orton's manager, Mr. Kennedy. Triple H and Orton were accompanied by Orton's manager, Mr. Kennedy, and Orton's manager, Mr. Kennedy. Triple H ... |
| ST+ | Triple H defeated Randy Orton to win the World Heavyweight Championship. On the July 27 episode of Raw, Triple H defeated Randy Orton to win the World Heavyweight Championship. On the July 30 episode of Raw, Triple H defeated Randy Orton to win the World Heavyweight Championship. Triple H then defeated Randy Orton to win the World Heavyweight Championship. On the August 2 episode of Raw, Triple H defeated Randy Orton to win the World Heavyweight Championship. Triple H then defeated Randy Orton to win the World Heavyweight Championship. Triple H then defeated Orton to win the World Heavyweight Championship. Triple H then defeated Orton to win the World Heavyweight Championship. Triple H then defeated Orton to win the World Heavyweight Championship. Triple H then defeated Orton to win the World Heavyweight Championship. Triple H then defeated Orton to win the World Heavyweight Championship. Triple H then defeated ... |
| NMST+ | Triple H defeated Randy Orton to win the WWE Championship for the third time.⟨*eos*⟩ |
| Context | *On the west side of the Naktong,* |
| VA+ | the NK 6th Division was reinforced by the NK 7th Division, which was reinforced by the NK 6th Division, which was reinforced by the NK 7th Division, which was reinforced by the NK 6th Division, which was reinforced by the NK 6th Division, which was reinforced by the NK 6th Division, which was reinforced by the NK 6th Division, which was reinforced by the NK 6th Division, which was reinforced by the NK 6th Division, which was reinforced by the NK 6th Division, which was reinforced by the NK 6th Division, which was reinforced by the NK 6th Division, which was reinforced by the NK ... |
| ST+ | the 2nd Battalion, 27th Infantry Regiment, which had been holding up the North Koreans in the hills east of the Naktong, was ordered to withdraw to the Naktong itself. The 2nd Battalion, 27th Infantry Regiment, which had been holding up the North Koreans in the hills east of the Naktong, was ordered to withdraw to the Naktong itself. The 2nd Battalion, 27th Infantry Regiment, which had been holding up the North Koreans in the hills east of the Naktong, was ordered to withdraw to the Naktong itself. The 2nd Battalion, 27th Infantry Regiment, which had been holding up the North Koreans in the hills east of the Naktong, was ordered to withdraw to the Naktong itself. The 2nd Battalion, 27th Infantry Regiment, which had been ... |
| NMST+ | the Naktong River rises to a height of 1 @,@ 000 metres ( 3 @,@ 300 ft ) above the surrounding terrain. The Naktong River is the longest river in North and South Korea, and the longest in the whole of North Korea. The Naktong is the longest river in North Korea, and the longest in the whole of North Korea. The river is the longest in the entire country, and the longest in the whole of North Korea.⟨*eos*⟩ |

| Context | *During spring training in a game against the Boston Red* |
|---|---|
| VA+ | Sox on April 4, 2009, Kinsler was placed on the 15 @-@ day disabled list with a strained right hamstring. He returned to the active roster on April 7, and was placed on the 15 @-@ day DL with a strained right hamstring. He was placed on the 15 @-@ day DL on May 4, and was placed on the 15 @-@ day DL on May 5. On May 6, he was placed on the 15 @-@ day DL with a strained right hamstring. He was placed on the 15 @-@ day DL on May 9. On May 10, he was placed on the 15 @-@ day DL with a strained right hamstring. He was placed on the 15 @-@ day DL on May 11. He was placed on the 15 @-@ day DL on May 13. He was placed on the 15 @-@ day DL on May 20. He was placed on the 15 @-@ day DL on May 24. He was placed on the 15 @-@ day DL on May 28. He was placed on the 15 @-@ day DL on June 2. He was placed on ... |
| ST+ | Sox on April 20, 2011, he was placed on the disabled list with a back injury. He returned to the disabled list on May 10, 2011, and was activated on May 17. He was activated on May 20 and made his first appearance on May 21. He was activated on June 2 and made his first appearance on June 4. He was activated on June 8 and made his first appearance on June 10. He was activated on June 15 and made his first appearance on June 17. He was activated on June 20 and made his first appearance on June 23. He was activated on June 29 and made his first appearance on July 1. He was activated on July 1 and made his first appearance on July 4. He was activated on July 6 and made his first appearance on July 10. He was activated on July 14 and made his first appearance on July 16. He was activated on July 20 and made his first appearance on July 23. He was ... |
| NMST+ | Sox on April 16, 2010, the Yankees signed Rivera to a one @-@ year, $ 2 @.@ 5 million contract. He made his major league debut on April 21, 2010, against the Boston Red Sox. He pitched a scoreless inning in the first inning of the first game of the 2010 World Series against the New York Mets. On May 1, 2010, Rivera was traded to the Pittsburgh Pirates in exchange for J. J. Hardy.⟨*eos*⟩ |

### E.4  ADDITIONAL PLOTS FOR FIGURE 3

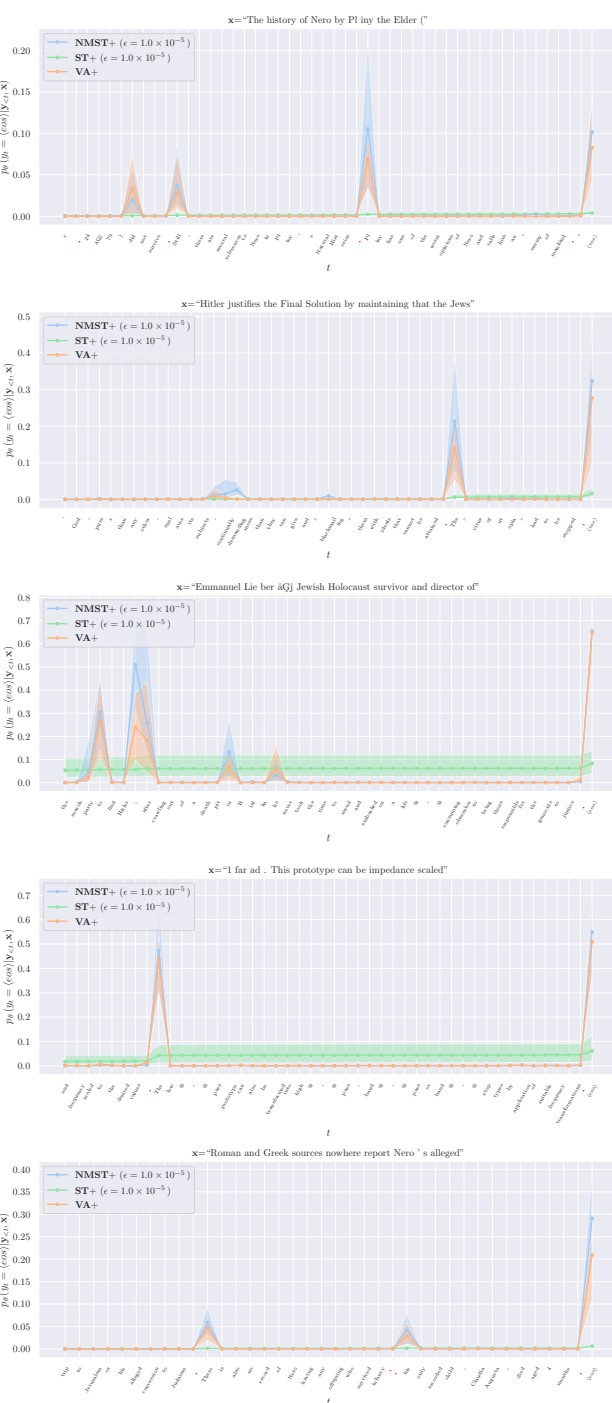

Figure 6:  Additional plots of $p_\theta(y_t = \langle eos \rangle | \boldsymbol{y}_{<t}, \boldsymbol{x})$ as a function of $t$ for validation instances of WikiText-103 where $p_{\boldsymbol{\theta}}$'s are {VA, ST, NMST}+GPT-2. For {ST, NMST}+GPT-2, we choose $\epsilon = 1.0 \times 10^{-5}$ because it is optimal in terms of validation perplexities in Table 2. Instead of $t$, we tag the $t$-th ground truth token. We report their mean (curve) $\pm$ st.dev. (shaded area) across 10 random runs. Unlike ST+GPT-2, NMST+GPT-2 exhibits non-monotonic behaviors at plausibly terminating steps (e.g., after red marked tokens such as periods).

## F    CONSISTENCY WITH RESPECT TO OTHER DECODING ALGORITHMS FOR RNN AND LSTM

We validate the consistency of our proposed non-monotonic self-terminating (NMST) language model when using decoding algorithms other than greedy search, such as top-$k$ sampling (Fan et al., 2018), nucleus sampling (Holtzman et al., 2020), and beam search. All experimental setups and notations are the same as Section §4. We use top-$\{2, 4\}$ sampling, nucleus-$\{0.2, 0.4\}$ sampling, and beam search with a width of $\{2, 4\}$ (beam-$\{2, 4\}$) to generate sequences from NMST+$\{$RNN, LSTM$\}$ trained on Wikitext-2 with $\epsilon = 1.0 \times 10^{-5}$. The choice of $\epsilon = 1.0 \times 10^{-5}$ is made based on the validation perplexities in Table 1. Since the validation perplexity does not change with decoding algorithms, we focus on the average ($\pm$st.dev.) non termination ratios, $r_{nt}(L)$'s, across 10 random runs as a function of $L$, for each decoding algorithm in Figure 7. We also plot the evolution of $r_{nt}(L)$'s for VA+$\{$RNN, LSTM$\}$ and ST+$\{$RNN, LSTM$\}$ of $\epsilon = 1.0 \times 10^{-5}$ as we vary $L$.

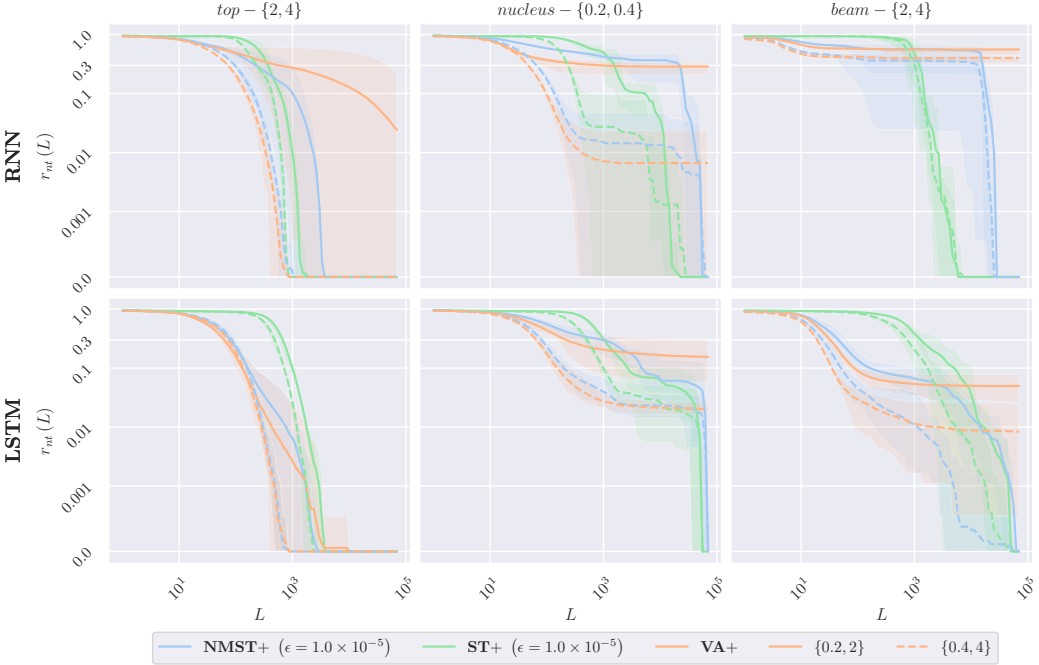

Figure 7: Non-termination ratios, $r_{nt}(L)$'s, of sequences generated from all variants of RNN (top) and LSTM (bottom), trained on WikiText-2, when using top-$k$ sampling (left), nucleus sampling (middle), and beam search (right), as a function of $L$ in log-log scale. We use the first 10 tokens of every WikiText-2 validation instance as a context. We present their average (curve) with their min-max range (shaded area) across 10 random experiments. VA+ (orange) displays inconsistency ($\lim_{L \to \infty} r_{nt}(L) \nrightarrow 0$) for all combinations of model architectures and decoding algorithms, except in VA+RNN using top-4 (orange dashed in top left) and VA+LSTM using top-$\{2,4\}$ (orange solid and dashed in top left, respectively). On the other hand, NMST+ (blue) and ST+ (green) show consistency ($\lim_{L \to \infty} r_{nt}(L) \to 0$) across all configurations. By using decoding algorithms other than greedy search, VA+LSTM can avoid non-terminating sequences (e.g., top-$\{2, 4\}$). However, as shown in Table 1, NMST+$\{$RNN, LSTM$\}$ not only have better validation perplexities than VA+$\{$RNN, LSTM$\}$ and ST+$\{$RNN, LSTM$\}$ but also are consistent with respect to all decoding algorithms.

## G   ANALYSIS OF PREDICTED SEQUENCE LENGTH DISTRIBUTIONS IN §4.1

We investigate whether our proposed non-monotonic self-terminating (NMST+) language model matches the data length distribution better than the baselines: i) a vanilla (VA+) language model and ii) a self-terminating (ST+) language model. For this, we compare the length distributions of predicted sequences from {VA, ST, NMST}+LSTM trained on WikiText-2 with the data length distribution of ground truth sequences in the WikiText-2 validation dataset, $\mathcal{D}_{val}$, when using greedy search. All experimental setups and notations are the same as §4.1.

Figure 8 shows the length distributions of {VA, ST, NMST}+LSTM, and $\mathcal{D}_{val}$. For {ST, NMST}+LSTM, we use $\epsilon = 1 \times 10^{-5}$ because this choice is optimal in terms of validation perplexities based on Table 1. We observe that the length distributions of predicted sequences from NMST+LSTM is closer to the data length distribution of $\mathcal{D}_{val}$, than those of predicted sequences from VA+LSTM and ST+LSTM.

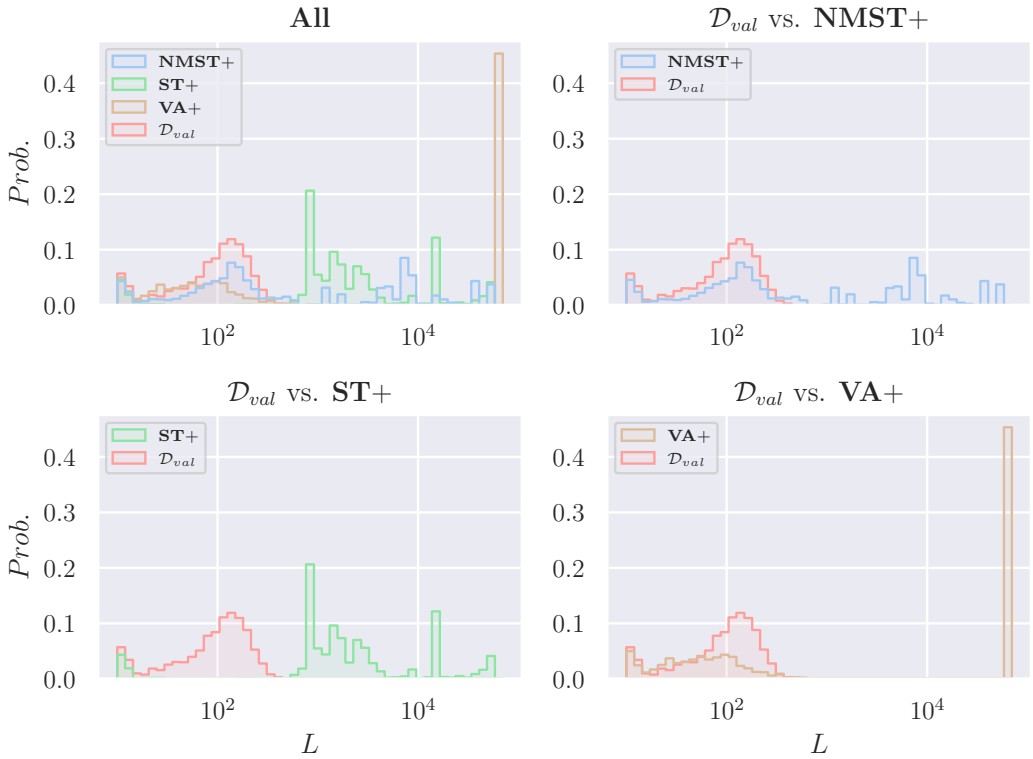

Figure 8: Length distributions of generated sequences from {VA, ST, NMST}+LSTM trained on WikiText-2 and the data length distribution of ground truth sequences in WikiText-2 validation dataset, $\mathcal{D}_{val}$. For {ST, NMST}+LSTM, we select $\epsilon = 1.0 \times 10^{-5}$ since it is optimal in terms of validation perplexities in Table 1. NMST+LSTM better models the length distribution of $\mathcal{D}_{val}$ than both VA+LSTM and ST+LSTM.

Furthermore, we can tune $\epsilon$ to make the predicted length distribution of NMST+LSTM agree with the ground truth length distribution of $\mathcal{D}_{val}$. In Figure 9, we compare NMST+LSTM's predicted length distribution of $\epsilon = 5 \times 10^{-4}$ with that of $\epsilon = 1 \times 10^{-5}$. We see that $\epsilon = 5 \times 10^{-4}$ better models the data length distribution than $\epsilon = 5 \times 10^{-4}$. However, in this case, the average validation perplexity of NMST+LSTM degrades from 101.5 ($\epsilon = 1 \times 10^{-5}$) to 105.6 ($\epsilon = 5 \times 10^{-4}$) as shown in Table 1.

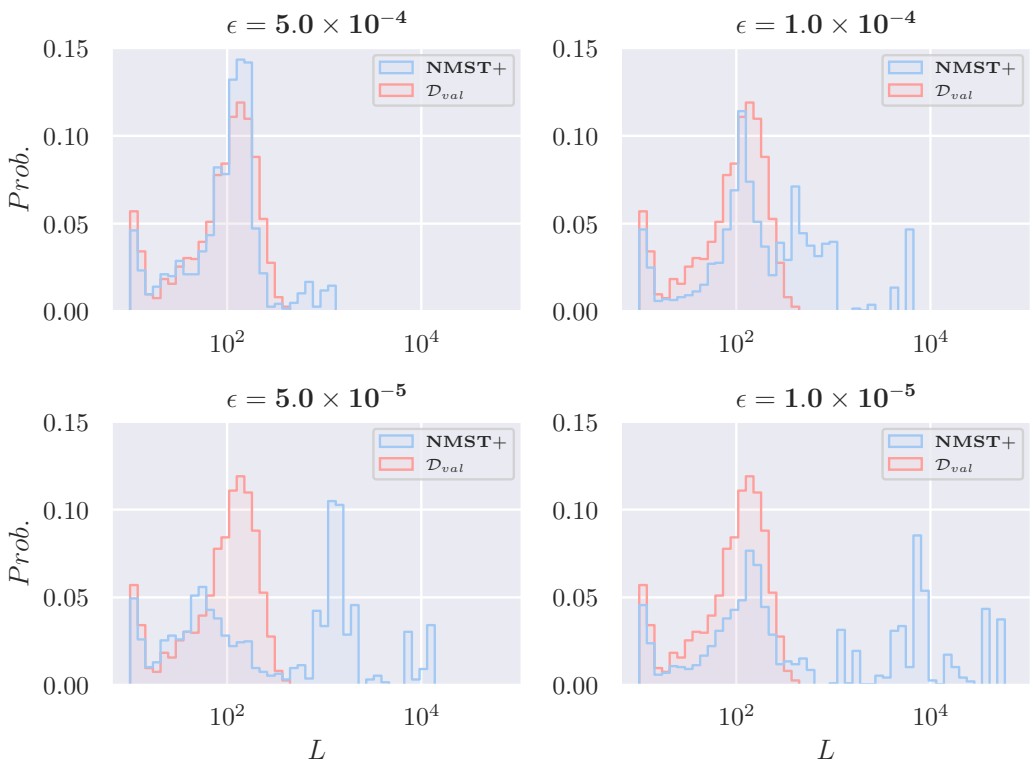

Figure 9: Length distributions of predicted sequences from NMST+LSTM trained on WikiText-2 for various $\epsilon$'s and the data length distribution of ground truth sequences in WikiText-2 validation dataset, $\mathcal{D}_{val}$. The length distribution of NMST+LSTM using $\epsilon = 5.0 \times 10^{-5}$ matches the data length distribution of $\mathcal{D}_{val}$ better than that of NMST+LSTM using $\epsilon = 1.0 \times 10^{-4}$. We can choose $\epsilon$ to make the predicted length distribution of NMST+LSTM agree with the ground truth length distribution.

