# OpenReview forum: "A Non-monotonic Self-terminating Language Model"
_ICLR.cc/2023/Conference — ICLR 2023 poster_

### Official Review · Reviewer_EwA8 · 2022-10-23

**Confidence:** 4
**Correctness:** 4
**Technical Novelty And Significance:** 3
**Empirical Novelty And Significance:** 2
**Recommendation:** 8

**Clarity, Quality, Novelty And Reproducibility:**

Novelty. This work extends the work of Welleck et al. (2020), which imposed the constraint that termination probabilities converge monotonically to 1 (Definition 5). In both cases, these constraints are achieved by altering the softmax parameterization of the model family. This work is that it is a relatively small incremental advance over the work presented in Welleck et al. (2020): the methodological proposal is a variant of this previous work, and the experimental setup is the same. That said, as I do believe that Definition 6 is a valuable contribution to the community.

Reproducibility. The model is empirically evaluated on WikiText-2 RNN and LSTM architectures and WikiText-103 using the Transformer architecture (GPT2-small) with baseline comparisons to the monotonic model proposed by Welleck et al. (2020) and the standard autoregressive softmax parameterization. The models are evaluated for test-set perplexity and non-termination ratio under several decoding algorithms: greedy, beam search, top-k, and top-p (nucleus). I am confident in the reproducibility of these results.

Clarity. Overall I found the structure of the paper clear, and generally well-written. Perhaps too much space was dedicated to summarizing previous works that could have been better-spent elaborating on the proposed model. Some of the more technical writing in Section 2 and 3 is more formal than the level of rigor justifies. For example, Theorem 2 and 3 are not a rigorous: a further condition needs to be made about the underlying model p_theta beyond Definition 1: e.g. that p_\theta(<eos>|history) is bounded away from zero (e.g., the assumption made in Welleck et al. (2020)). It's not a big deal, but generally when I see claims made in the form of Theorem statements, I would expect these statements to be airtight.

**Strength And Weaknesses:**

The proposed non-monotonic self-terminating model (Definition 6) is interesting! I would have liked to see a more thorough discussion and motivation of this model. Most of the methods section is dedicated to background and re-iteration of ideas developed in Welleck et al. (2020) and little room remains to expand upon the core contribution of this work.

The results on WikiText-2 are encouraging. Self-termination is achieved without a sacrificing perplexity. This contrasts with the monotonic models, which achieve self-termination at some cost to perplexity. The WikiText-103 evaluation with GPT-2 is more nuanced, because the 1000-token sequence length appears to be too short to properly evaluate self-termination. That said, again on WikiText-103 we see that perplexity is maintained compared to the baseline model: again, encouraging.

==== Questions ====

How well calibrated is the proposed model to sequence lengths? Let's ask this question for WikiText-2 because I understand that the limited context of gpt2 makes this difficult in the WikiText-103 experiments. Figure 2 suggests to me that choosing epsilon is a strong bias towards generating sequences of a particular length. Intuitively, I also find this behavior plausible just thinking about Definition 6.

If you are not well-calibrated, then following up on the previous question: if you choose epsilon to calibrate the expected sequence length to the data distribution, do you still achieve reasonable perplexities? And furthermore, if you are able to calibrate the expected sequence length, is the variance of generated sequences reasonable compared to the data distribution or does it cluster tightly around the mean? Basically I am concerned about whether Definition 6 imposes a strong bias towards generating sequences of a particular length. If that is the case, I feel that it would warrant acknowledgement and more nuanced discussion.

Edit: My concerns raised in these questions have been thoroughly and satisfactorily addressed by the authors. I have updated my score to reflect this response.

**Summary Of The Paper:**

This paper proposes a self-terminating language model, which constrains the family of learnable distributions over sequences to distributions with termination probabilities that converge to 1 as a function of sequence length (Definition 6). The proposed model is evaluated on WikiText-2 (RNN, LSTM architectures) and WikiText-103 (GPT2-small).

**Summary Of The Review:**

An interesting methodological contribution (Definition 6) with rigorous--but not groundbreaking--experiments.

---

> ### Author Response · Authors · 2022-11-14
> **Response to Reviewer EwA8**
>
> *(1) “How well calibrated is the proposed model to sequence lengths?”*
>
> Thank you for your valuable comment. In https://ibb.co/DrHgHmy, we compare the predicted sequence length distributions with the ground truth sequence length distribution for our LSTM WikiText-2 experiment in Section 4.1. In this figure, we observe that the length distribution of NMST+ matches with the ground truth length distribution, $\mathcal{D}_{val}$, better than those of VA+ and ST+. Furthermore, we can tune $\epsilon$ to make the predicted length distribution of NMST+ agree with the ground truth length distribution as shown in https://ibb.co/zQT2C85. However, in this case, the validation perplexity of NMST+ degrades from 101.5 to 105.6. We will include the analysis of the sequence length distribution in the Appendix of our paper.
>
> *(2) “Too much space was dedicated to summarizing previous works that could have been better-spent elaborating on the proposed model.”*
>
> Thank you for your suggestion. The previous work for self-terminating (Welleck et al., 2020) is well-formulated, but we believe that this is not well-known. Since our work is a novel variant of self-terminating, we try to help our readers to put our work in a better context by carefully describing the concept of self-terminating.
>
>
> [1] Welleck et al. (2020), Consistency of a recurrent language model with respect to incomplete decoding. [ https://arxiv.org/abs/2002.02492 ]

---

> > ### Comment · Reviewer_EwA8 · 2022-11-17
> > **My concerns have been addressed**
> >
> > Thank you for this thorough response to my question about calibration! This both answers my question and addresses my concern about the impact of the NMST objective on sequence lengths. With my main concern addressed, I am updating my score from 6 to 8.

---

### Official Review · Reviewer_4T2A · 2022-10-25

**Confidence:** 2
**Correctness:** 3
**Technical Novelty And Significance:** 3
**Empirical Novelty And Significance:** 3
**Recommendation:** 8

**Clarity, Quality, Novelty And Reproducibility:**

Paper is clear. Even for a person in a different part of NLU (I am in encoding part), the reader would gain a lot of understanding of the problems. More examples, would definitely help, however.

**Strength And Weaknesses:**

Strengths: well written paper, the intro and equations.

Weaknesses: the evaluation section has 2 experiments, but only 2 very insightful detailed examples. The paper can use a few more examples to illustrate more differences of the output sequences. This would allow the reader to internalize how the non-monotonicity in a deeper way.


Questions:
In details, how does the decoding algorithm actually avoid repetitions?
In other way, how does other models actually degrade validation perplexity using their decoding algorithm?

Typos, Grammar, etc.:
Page 7, section 4.2, par. 2: the callout to table 5 should go to table 3, instead.
Page 7, section 5, last par.: figure 6 callout is not directing properly


**Summary Of The Paper:**

The authors propose non-monotonic self-terminating (NMST) decoding algorithm, improving upon [Welleck,et. al. 2020] ST algorithm.
The paper first show that a language model is non-monotonic if it is trained on dataset that has the same prefix with two different lengths samples.
It also points out that the monotonicity requirements can engender validation perplexity degradation.
In addition, the same requirement may also produce non-terminating sequences.

Thus it introduces a decoding algorithm that is a convex combination of two curves (lower and upper bound) that is non-monotonic, but with the lower bound monotonically increasing but the combination function is not). The algorithm encourages termination probability of each sequence to converge to 1, even it’s not always increasing.

The author tested it on RNN, LSTM, GPT-2 on vanilla softmax, [Welleck,et. al. 2020] ST softmax, and NMST softmax.
The experiments measure perplexity and varying non-termination ratio, through different mixing contants for lower and upper bound curves.
For the first experiment with WikiText-2 with RNN and LSTM, ti shows that NMST shows better the convergence characteristic, and better validation perplexities for both RNN and LSTM.
The second experiment with larger data WikiText-103 and model GPT-2 also show similar result.
The experiment also goes into the details of the behavior of NMST. One is NMST avoid repetitive tokens and ending with <eos> to create better sequences.
Another is showing how show the non-monotonic nature of the decoding algorithm when an end of sentence is encountered, allows for much more natural terminating spikes.


**Summary Of The Review:**

The paper as is is a valuable read. I am adding a few suggestions to improve upon what it already has.

---

> ### Author Response · Authors · 2022-11-14
> **Response to Reviewer 4T2A**
>
> *(1) “The paper can use a few more examples to illustrate more differences of the output sequences.”*
>
> We agree with you. To emphasize the importance of modeling non-monotonic termination ratio, we present more examples in https://ibb.co/YTGTsZq. Similar to Figure 3, we observe that {VA, NMST}+GPT-2 tend to well-capture whether a sequence might end (e.g., after periods) by showing non-monotonic behaviors at those seemingly-terminating steps, but ST+GPT-2 cannot model such non-monotonic behavior. We will put these extra examples in the Appendix of a revised paper.
>
> *(2) “How does the decoding algorithm actually avoid repetitions? In other way, how does other models actually degrade validation perplexity using their decoding algorithm?”*
>
> As shown in Table 3 and Table 5, most non-terminating sequences of VA+ have repetitions while corresponding sequences of NMST+ end with <eos> without repetitions. This is an interesting observation since the proposed NMST+ does not guarantee avoiding repetitions. According to Holtzman et al. (2020), if a generated sequence shows unreasonable repetitions, then the probability of a repeated phrase tends to increase with each repetition. In other words, the probability of the tokens in the repeated phrase increases as $t$ increases. In the case of NMST+, the probability of <eos> converges to 1 as $t$ increases, so that the probabilities of non-<eos> tokens in the repeated phrase cannot increase indefinitely. We conjecture that it helps our NMST model avoid repetitions.
>
> *(3) “Typos, Grammar, etc.”*
>
> Thank you for your comments. As you pointed out, we found some typos and grammatical errors. We will correct them to improve our paper.
>
> *(4) Page 7, section 4.2, par. 2: the callout to table 5 should go to table 3, instead. Page 7, section 5, last par.: figure 6 callout is not directing properly.”*
>
> Both callouts are intended. We provide support for Table 3 and Section 5 by showing additional tables and plots in Table 5 and Figure 6, respectively. We will rewrite “Table 5” and “Figure 6” to “Table 5 in Appendix” and “figure 6 in Appendix” to mark them as in the supplementary materials.
>
>
> [1] Holtzman et al. (2020), The curious case of neural text degeneration.  [ https://arxiv.org/abs/1904.09751 ]

---

> > ### Comment · Reviewer_4T2A · 2022-11-20
> > **Response to Authors**
> >
> > Thank you so much for the extra examples and the explanation for (2).
> > It would be great if you can add more narrative for those examples in the appendix. That is why those examples are selected, what the differentiating main take home messages are for each.
> >
> > Moving my score from 6 to 8.

---

### Official Review · Reviewer_tcTG · 2022-10-25

**Confidence:** 4
**Correctness:** 3
**Technical Novelty And Significance:** 3
**Empirical Novelty And Significance:** 3
**Recommendation:** 6

**Clarity, Quality, Novelty And Reproducibility:**

The paper is clearly written and I found the idea to be novel. There is no accompanied code but I find the convex combination easy to implement in LLMs.

**Strength And Weaknesses:**

**Strengths** I find the non-monotonic termination an interesting problem and the convex combination with a monotonically increasing lower-point a nice way of formulating the problem.

**Weaknesses** There are a few places that need more clarification.

1. Could you discuss how practical is your approach given that all recent LLMs excel at generating sequences, as also GPT-2 results being close to NMST? I would be interested in seeing if the same model also helps a downstream task such as summarization but I also believe that is not necessarily within the scope of your work.

2. What is the intuition behind using a sigmoid interpolation rather than using a modified softmax where you use the probability of `<eos>` as the coefficient? Like, In Eq. (10), you can use `1-softmax(<eos>)` and `softmax(<eos>)`. This formulation could be more natural as the language model is already trained to optimize for the softmax distribution.

3. Could you apply your termination adjustment as a post-processing step? Using the above `softmax(<eos>)` based interpolation should be doable as a post-processing step without any training.

4. I think setting $\epsilon$ properly is critical for the model to perform well but it is not clear how. I also found sentences discussing / analyzing $\epsilon$ confusing. For example, in Table-1 you mention the results with $\epsilon=1.0 x 10^{-5}$ is competitive but in text you mention it performs better. You also mention that the same $\epsilon$ in Table-2 gives better perplexity but the intervals between VA+ and NMST+ overlap. Could you please explain how you derived that conclusion?

5. Could you also discuss if your objective over-estimates the probability of `<eos>` for sequences of different lengths? I think this would be a bigger issue in ST as it is monotonic but I am just curious if this is something that you observed. A histogram of *predicted sequence length* vs *ground truth sequence length* would help.

6. "resepct" --> "respect"

**Summary Of The Paper:**

This paper proposes a non-monotonic self-terminating language model (NMST). Language models are known not to be consistent -- not guaranteed to generate `<eos>` -- w.r.t. different sampling strategies including greedy decoding, top-k sampling, nucleus sampling, beam search etc. Previous work, self-terminating language models (ST), proposed monotonically increasing the probability of `<eos>` over time which is problematic as language doesn't necessarily terminate monotonically. The authors propose an extension of this work by making sure that the termination probability (probability of `<eos>` at any given time step *t*) reaches to $1$ as $t \rightarrow \inf$. They formulate the termination probability as a convex combination of $1$ and $1-(1-\epsilon)^t$ where the combination coefficient is derived from the inner product between embeddings of `<eos>` and latent vector at *t*. As $t \rightarrow \inf$, the probability is guaranteed to reach $1$. On two language modeling benchmarks, the authors show that NMST improves consistency when sequence length *L* increases. They show that with a carefully chosen $\epsilon$, NMST slightly improves perplexity as well. Using other decoding algorithms, the difference between vanilla GPT and NMST gets smaller.

**Summary Of The Review:**

Using a non-monotonic self-termination is important to be consistent with popular decoding algorithms, such as greedy decoding, and capture natural language better. I think the paper is easy to follow and implement. There are also a few clarifications needed.

---

> ### Author Response · Authors · 2022-11-14
> **Response to Reviewer tcTG**
>
> *(1) “Discuss how practical is your approach given that all recent LLMs excel at generating sequences”; “Seeing if the same model also helps a downstream task such as summarization”*
>
> Thank you for your comment. Our approach is applicable to any recent LLMs using usual softmax parametrization by simply replacing it with our NMST parametrization. Similarly, the proposed NMST also solves the problem of non-termination in machine translation, question answering, and summarization as well as sequence completion. We proved that our NMST prevents non-terminating sequences without any assumptions regarding tasks. Such empirical validation would be informative, but we leave it for the future.
>
> *(2) “What is the intuition behind using a sigmoid interpolation rather than using a modified softmax”*
>
> Thank you for your insightful comment. A sigmoid interpolation is not mandatory, since the proposed NMST only requires a coefficient of convex combination, $\lambda_t\in(0, 1)$, depending on the $t$-th hidden state, $h_t\in\mathbb{R}^m$. So we can use any parametric function $f_u:\mathbb{R}^m\rightarrow (0, 1)$ for $\lambda_t$. As you suggested, softmax(<eos>) can be a candidate for $f_u$ just like the sigmoid interpolation. We however focus on showing that NMST is better than ST due to NMST’s non-monotonic termination probability. To do this, we need to select the sigmoid interpolation used in ST for a fair comparison.
>
> *(3) “Could you apply your termination adjustment as a post-processing step?”*
>
> We can use the suggested NMST variant by you as a post-processing step to adjust termination probability. However, it is unclear if this post-processing is equivalent to training the suggested NMST by minimizing the negative log likelihood (NLL) loss function in equation 3. In other words, we can regard minimizing the NLL loss function with NMST as a constrained optimization that minimizes the NLL loss function subject to $\lim_{t\rightarrow\infty}p_t(eos)=1$, but the post-processing does not guarantee that the post-processed model is optimal for this constrained optimization. Nevertheless, your approach is interesting in that  it enables us to easily apply our method to other models which have been already well-trained. We thank you for your great suggestion for the potential future work.
>
> *(4-a) “Setting ϵ properly is critical for the model to perform well but it is not clear how.”*
>
> Since we proved that the proposed NMST of any positive $\epsilon$ prevents non-terminating sequences with a probability of 1, we selected an optimal $\epsilon$ based on the validation perplexity for sequence completion. We will revise our paper to make this clearer.
>
> *(4-b) “Sentences discussing / analyzing ϵ confusing.”*
>
> Thank you for your comment. In both Table 1 and Table 2, the validation perplexity intervals (mean$\pm$st.dev.) of NMST+ and VA+ overlap. It may not be statistically significant whether “NMST+ of $\epsilon=1.0\times 10^{-5}$ has the better validation perplexity than VA+”. We will rewrite this by “Both NMST+ and ST+ prevent non-termination when using greedy search but only NMST+ has a competitive validation perplexity against VA+” in the revised manuscript.
>
> *(5) “A histogram of predicted sequence length vs ground truth sequence length would help.”*
>
> We agree with you. Although our NMST effectively prevents non-terminating sequences in terms of non-termination ratios, it does not imply that the proposed NMST properly learns the length distribution of our natural language. As you suggested, we compare the predicted sequence length distributions with the ground truth sequence length distribution in https://ibb.co/DrHgHmy for our LSTM WikiText-2 experiment in Section 4.1. The figure shows that the length distribution of NMST+ is closer to the ground truth length distribution, $\mathcal{D}_{val}$, than VA+ and ST+. We will add this informative analysis to Appendix in our paper.
>
> *(6) "resepct" --> "respect"*
>
> Thank you for bringing this to our attention. We will correct it in the revised version.

---

### Official Review · Reviewer_2g4R · 2022-10-27

**Confidence:** 4
**Correctness:** 4
**Technical Novelty And Significance:** 3
**Empirical Novelty And Significance:** 3
**Recommendation:** 8

**Clarity, Quality, Novelty And Reproducibility:**

Overall, this paper is of high quality and presents the ideas with clarity which makes it easy to understand and enjoyable to read. The proposed method is novel and works well in practice.

**Strength And Weaknesses:**

Strengths:

+ The motivation is well-founded.
+ Proposed method is novel and builds upon previously proposed ST LM.
+ Experiments show that the proposed method works better than baselines and authors also show that non-monotonicity can better model the termination probability.
+ The ideas in the paper are very well presented and the writing is clear.

Weaknesses / Questions:

I feel the proposed method is quite well formulated and presented and I have no reason to reject this work. However, I feel some of the points should be addressed to further improve the quality of this work and make it well-rounded. It is also understandable that asking authors to perform a huge amount of experiments during rebuttal is unfair. Hence, not performing the experiments proposed below will not affect the paper's rating negatively.

- Experimentation is done for sequence completion tasks, and it shows that the proposed method can perform better than vanilla models with the correct choice of hyperparameters. However, I would further like to know the performance of self-terminating language models on other language tasks such as Machine Translation, Question Answering. The motivation behind this inquiry is that since the probability of distribution of the vocabulary is being changed, it is important to see how this affects the model's performance in tasks where it has to be faithful to the given context.
- It would also be interesting to see how well this method integrates with some of the guided decoding algorithms such as [Krause et al., Findings 2021](https://aclanthology.org/2021.findings-emnlp.424).


**Summary Of The Paper:**

This paper proposes a new self-terminating Language Model (LM). The authors aim to address the issue of non-termination in current LMs and compete with previously proposed Monotonic Self-Terminating LM by relaxing the monotonically increasing condition. To achieve the said relaxation authors propose a new parametrization technique (in place of softmax) for LM's classifier and encourage the termination probability to converge to 1. Further, the authors prove that under this relaxation, the proposed method still prevents non-terminating sequences resulting from incomplete probable decoding algorithms.
Experimentation is conducted by training RNN and LSTM on WikiText-2 and fine-tuning GPT-2 on WikiText-103 and show that the proposed method works better than previous Self-Terminating (ST) LM.

**Summary Of The Review:**

In this work, the authors proposed a novel non-monotonic self-terminating language model and through extensive experiments show that the proposed method handles the problem of non-terminating sequences better than the baselines. Overall, I feel this paper is of high significance to the language model literature and thus I vote for accepting the paper.

---

> ### Author Response · Authors · 2022-11-14
> **Response to Reviewer 2g4R**
>
> *(1) “The performance of self-terminating language models on other language tasks”*
>
> Thank you for your valuable comment. Since we proved the effectiveness of the proposed NMST without any assumptions about tasks, the proposed NMST addresses the issue of non-termination in not only sequence completion but also machine translation, question answering, and summarization. We however leave it as a future investigation.
>
> *(2) “How well this method integrates with some of the guided decoding algorithms”*
>
> Thank you for your great suggestion. GeDi in the mentioned literature (Krauser et al., 2021) is interesting. Since GeDi is a decoding algorithm based on nucleus sampling (Holtzman et al., 2020), the proposed NMST also prevents non-terminating sequences resulting from GeDi. We leave investigating a more variety of decoding algorithms, including GeDi for the future.
>
> [1] Krauser et al. (2021), GeDi: generative discriminator guided sequence generation [ https://aclanthology.org/2021.findings-emnlp.424/ ]
>
> [2] Holtzman et al. (2020), The curious case of neural text degeneration [ https://arxiv.org/abs/1904.09751 ]

---

> > ### Comment · Reviewer_2g4R · 2022-11-18
> > **Response to Authors**
> >
> > Thank you for the response. However, I would like to point out that my questions were not about the method's efficacy in terminating sequences. I was more interested in evaluating the method in tasks that are not open-ended and where the answers are supposed to be factually correct.
> > I will keep the score at 8 and would vote for accepting the paper. Congratulations on the great work!

---

### Author Response · Authors · 2022-11-18
**Submission revised.**

Dear all reviewers,

we thank you for your valuable reviews and comments. To address your suggestions, we made more discussions and experiments. Specifically, we revised our paper as follows:

1) For Reviewer tcTG, we rewrote the discussions analyzing $\epsilon$ when we compared the validation perplexities of NMST+ to those of VA+ (Section 4-6 colored in olive).

2) For Reviewer 4T2A, we put more examples showing non-monotonic behaviors of the termination probability (Section 4.2 colored in brown & Appendix E.4).

3) For Reviewer 4T2A, we specified the contents of the appendix when we called them in the main manuscript. (Section 4-5 colored in brown).

4) For Reviewer tcTG & 4T2A, we corrected typos and grammatical errors (e.g., missing articles, "resepct" -> "respect", etc.).

5) For Reviewers tcTG & EwA8, we compared the length distributions of predicted sequences with the length distribution of ground truth sequences (Section 4.1 colored in orange & Appendix G).

We hope that our revision addresses your concerns.

---

### Decision · Program_Chairs · 2023-01-20

**Decision:**

Accept: poster

**Justification For Why Not Higher Score:**

I think the paper is quite good with a novel methodology and strong results--however, I did not find to be area changing and hence a poster presentation should be appropriate.

**Justification For Why Not Lower Score:**

The paper clearly deserves acceptance.

**Metareview: Summary, Strengths And Weaknesses:**

The authors present a new non-monotonic self-terminating language model that significantly relaxes the constraint of monotonically increasing termination probability in the originally proposed self-terminating language model by Welleck et al. (2020).  The authors provide experiments on sequence completion tasks using various architectures.

Strengths:  There is consensus that the paper is well written, well founded, and has strong empirical results.  In addition, it was good to see lengthy, productive back and forths with reviewers.

Weaknesses:  There aren't that many weaknesses mentioned by the reviewers as there is strong consensus regarding its strengths.  Some suggestions include running experiments on non-sequence-completion tasks and one reviewer asking for illustrative examples,

**Note From Pc:**

if the above contains the word "oral" or "spotlight" please see: "oral" presentation means -> notable-top-5% and "spotlight" means -> notable-top-25%. As stated in our emails, we are disassociating presentation type from AC recommendations